



# Calibrating a large-domain land/hydrology process model in the age of AI: the SUMMA CAMELS experiments

Mozhgan A. Farahani[1], Andrew W. Wood[1,2], Guoqiang Tang[1], and Naoki Mizukami[3]

[1]Climate and Global Dynamics, National Center for Atmospheric Research, Boulder, CO, USA
[2]Civil and Environmental Engineering, Colorado School of Mines, Golden, CO, USA

[3]Research Applications Laboratory, National Center for Atmospheric Research, Boulder, CO, USA

*Correspondence to*: Mozhgan A. Farahani ([mozhgana@ucar.edu](mailto:mozhgana@ucar.edu)); Andrew W. Wood ([andywood@ucar.edu](mailto:andywood@ucar.edu))

**Abstract.** Process-based hydrological modeling is a long-standing strategy for simulating and predicting complex water processes over large, hydro-climatically diverse domains, yet model parameter estimation (calibration) remains a persistent
challenge for large-scale applications. New techniques and concepts arising in the artificial intelligence (AI) context for hydrology point to new opportunities to tackle this problem in process-based models. This study presents a machine learning (ML) based calibration strategy for large-domain modeling, implemented using the Structure for Unifying Multiple Modeling Alternatives (SUMMA) land/hydrology model coupled with the mizuRoute channel routing model. We explore various ML methods to develop and evaluate a model emulation and parameter estimation scheme, applied here to optimizing SUMMA
parameters for streamflow simulation. Leveraging a large-sample catchment dataset, the large-sample emulator (LSE) approach integrates static catchment attributes, model parameters, and performance metrics, providing a basis for large-domain regionalization to unseen watersheds. The LSE approach is compared with a single-site emulator (SSE), demonstrating improved calibration outcomes across temporal and spatial cross-validation experiments. The joint training of the LSE framework yields comparable performance to traditional individual basin calibration while enabling potential for parameter
regionalization to out-of-sample, unseen catchments. Motivated by the need to optimize complex hydrology models over continental-scale domains to support national water security applications, this work introduces a scalable strategy for the calibration of large-domain process-based hydrological models.

## 1 Introduction

Hydrological modelings advances have significantly expanded our capacity to simulate and predict complex water-related
processes. Such models provide critical information for water resource management and planning, flood hazard prevention, and climate resilience studies, among other applications. Accurate hydrologic simulations are vital in regions as expansive and diverse as the contiguous United States (CONUS), if not the globe, where variability in climate, land cover, and hydrological responses can be a challenge for the seamless implementation of land/hydrology models (LHMs: i.e., hydrologic models and/or the hydrologic components of land models). Traditional calibration approaches that involve tuning model parameters for



individual basins can be time-intensive, spatially non-generalizable and computationally costly, which limits their suitability
for large-domain (national, continental, global) applications. Because parameter estimation is vulnerable to sampling and input
uncertainty and input errors, such basin-specific methods often lead to spatial inconsistencies in parameter estimates, limiting
the model's generalizability across broader regions.

Recent advances and applications in artificial intelligence (AI) -- a family of methods including machine learning (ML), deep
learning (DL), large language models (LLMs) and other methods -- have been demonstrated to provide not only a skillful
strategy for simulating hydrology (Kratzert et al., 2024; Nearing et al., 2024; Arsenault et al., 2023; Feng et al., 2020), but also
for process-based (PB) hydrology model calibration. Calibration methods in hydrology are numerous and have a long history,
advancing hand-in-hand with the proliferation of models ranging in complexity from low-dimensional conceptual schemes
(commonly used in engineering applications and operational forecasting) to more explicit high resolution physics-based
models used in watershed and Earth System science. The greater complexity of such models drove calibration method
innovations such surrogate modeling for individual basins (Gong et al., 2016; Adams et al., 2023), which enabled a less-costly
interrogation of the model parameter space despite the models' increased computational demand. Such techniques have even
more recently been discovered by the Earth System modeling (ESM) community, which previously calibrated complex ESM
components (e.g., ocean, atmosphere, land) through ad hoc manual parameter tuning and sensitivity tests. AI-based methods
including model emulators are now increasingly used for exploring parameter related uncertainty and constraining model
implementations (Dagon et al., 2020; Watson-Parris et al., 2021; Bennett et al., 2024).

ML hydrology modeling applications have yielded the remarkable (and perhaps in retrospect, unsurprising) finding that joint
model training across many watersheds can learn robust, heterogeneous hydrometeorological relationships that enable them to
predict hydrological behavior for unseen watersheds and time periods -- which represents a large step forward in solving the
longstanding hydrological prediction-in-ungauged-basins challenge (PUB; Wagener et al., 2007; Hrachowitz et al., 2013). Mai
et al. (2022) clearly demonstrated the superior performance of Long Short-Term Memory (LSTM) networks in out-of-sample
temporal and spatial hydrologic simulation compared to a range of results from non-ML models. Such regionalization ability
had not been achieved previously with conceptual and PB hydrology models, where joint multi-site or regional training more
often comes at a cost to individual basin model performance (Mizukami et al., 2017; Samaniego et al., 2010; Tsai et al., 2021;
Kratzert et al., 2024), notwithstanding some gains in regional parameter coherence. Samaniego et al. (2010) achieved moderate
success at parameter regionalization using a joint large-domain training solution involving calibrating the coefficients of
transfer functions relating geophysical attributes ('geo-attributes') to model parameters, expanding on common pedotransfer
concepts for soil parameters. Since then, ML and DL approaches, including differentiable modeling -- e.g., embedding of DL
elements within conceptual models converted to differentiable form (Feng et al., 2020; Shen et al., 2023) -- and hybrid
ML/conceptual models (Frame et al., 2022) have continued to advance, outperforming traditional models and showing new
potential for generalizing to ungauged basins with diverse hydroclimatic conditions (Kratzert et al., 2024; Feng et al., 2020).



The aim of the research described in this paper is to surmount traditional basin-specific calibration challenges by leveraging insights from recent AI-related progress in hydrology. The specific objective (and research sponsor motivation) for the study is to calibrate a physics-based PB LHM, the Structure for Unifying Multiple Modeling Alternatives (SUMMA; see Sect. 2.2) over the entire CONUS for use in generating a large ensemble of future climate-informed hydrologic scenarios for use by US federal water agencies and others in water security applications -- e.g., agency guidance and long-term planning studies. Prior experience with individual basin calibration followed by regionalization, and associated performance limitations, motivated this investigation of possibilities for a more scalable approach.

To this end, we present an ML-based model calibration and regionalization strategy and associated method evaluation experiments for the CONUS-wide implementation of SUMMA, which is also demonstrated for calibrating the hydrology component of a land model in a companion paper by Tang et al. (2024). The approach employs a combined model emulation and parameter optimization scheme to estimate parameters jointly across diverse catchments, building from recent advances in the ML hydrologic modeling community. By training an emulator on a large sample catchment dataset to predict model performance as a function of catchment geo-attributes and parameters, we build the capacity for identifying optimal parameter sets across large, varied and unseen domains. The following sections describe the methods and results of a series of experiments with this approach as applied to a large collection of US watersheds, followed by discussion and conclusions.

## 2 Methods

### 2.1 Study domain

The study focuses on CONUS, a region encompassing diverse hydrological conditions due to its varied climate, landforms, and vegetation types. To represent such variability, we utilize a subset of watersheds from the Catchment Attributes and Meteorology for Large-sample Studies (CAMELS) dataset, which combines static catchment attributes with hydrometeorological time series for use in benchmarking hydrological modeling applications (Newman et al., 2015; Addor et al., 2017). Such datasets are well-suited for large-scale modeling due to their rich suite of attributes, including climate indices, soil properties, land cover, and streamflow observations, which provide a comprehensive basis for model calibration, evaluation and regionalization over a diverse range of hydroclimate settings. We selected 627 headwater basins from the 671 CAMELS basins, excluding those with nested interior or upstream areas to ensure independence and avoid overlapping drainage areas. Catchment boundaries for the modeling were updated from those provided in the original CAMELS dataset, correcting inaccuracies in boundary and drainage areas by using the original boundaries from the Geospatial Attributes of Gages for Evaluating Streamflow, version II dataset (Falcone, 2011), which are consistent with U.S. Geological Survey (USGS)-estimated drainage areas.



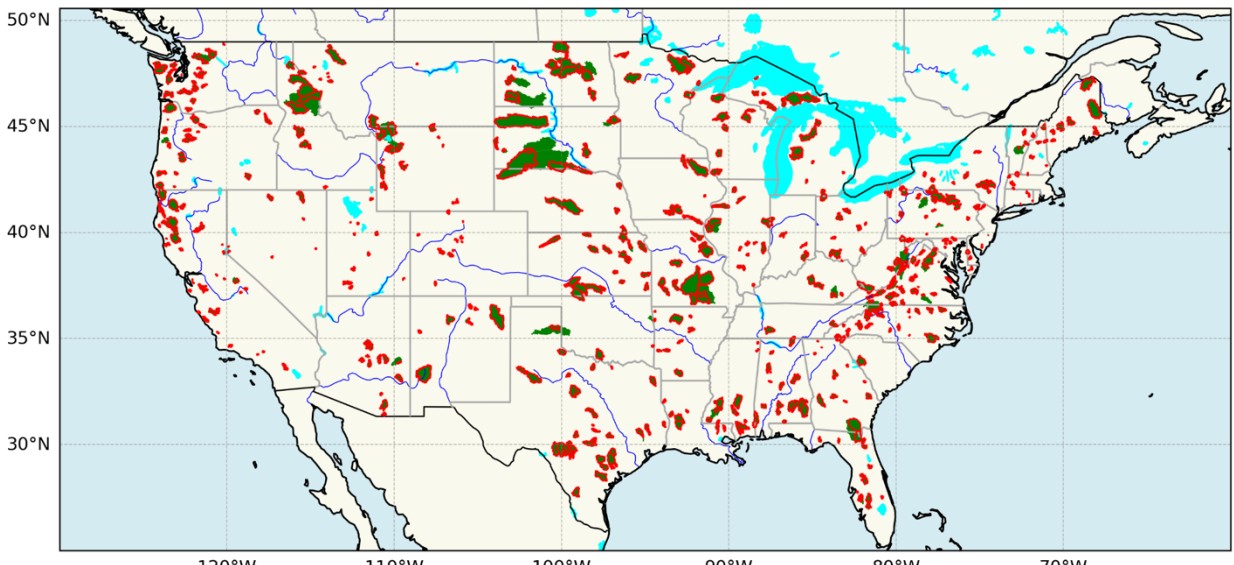

**Figure 1. Spatial distribution of selected headwater basins (red outlines) from the CAMELS dataset (green areas) across CONUS.**

## 2.2 Process-based modeling with SUMMA and mizuRoute

The Structure for Unifying Multiple Modeling Alternatives (SUMMA) is a process-based LHM framework designed for flexibility in representing hydrological processes across diverse catchments (Clark et al., 2015a, 2015b, 2021). SUMMA solves generalized mass and energy conservation equations, offering multiple parameterization schemes for hydrological fluxes, and enabling flexible advanced numerical techniques to optimize solution performance. SUMMA represents watersheds with a hierarchical spatial organization centering on Grouped Response Units (GRUs) that are divisible into one or multiple

Hydrologic Response Units (HRUs). GRU geometry is user-defined and has varied in usage from mesoscale catchment boundaries to fine or mesoscale resolution grid, as well as point-scale simulations. Such configurations allow SUMMA to represent the natural topography of the domain to the extent warranted by a given application, thereby improving the interpretability of model results (Gharari et al., 2020).

Here as in other SUMMA modeling studies, SUMMA runoff and subsurface discharge are subsequently input to the mizuRoute

channel routing model (Mizukami et al., 2016), a flexible framework supporting multiple hydrologic routing methods to provide streamflow estimates at gage locations. MizuRoute organizes the routing domain using catchment-linked HRUs connected by stream segments (Mizukami et al., 2016, 2021). Currently five methods are offered in MizuRoute, of which the Diffusive Wave (DW) routing scheme, as implemented by Cortés-Salazar et al. (2023), was adopted here. Both SUMMA and mizuRoute model codes are open source and have been extensively sponsored by the US water agencies (the Bureau of



Reclamation and US Army Corps of Engineers, USACE) with growing support from other agencies in the US and internationally.

For this study, SUMMA and mizuRoute are run at a nominal 3-hourly simulation timestep, and the associated sub-daily forcing include precipitation, temperature, specific humidity, shortwave and longwave radiation, wind speed, and air pressure. Precipitation and temperature forcings were derived from the Ensemble Meteorological Dataset for Planet Earth (EM-Earth),
which provides 0.1° spatial and hourly temporal resolution, merging ground-station data with reanalysis for enhanced accuracy (Tang et al., 2022). EM-Earth integrates gap-filled ground station data with reanalysis estimates, offering improved accuracy over standalone reanalysis products. Wind speed, air pressure, shortwave and longwave radiation, and specific humidity inputs were derived from ERA5-Land, also at 0.1° spatial and hourly temporal resolutions (Muñoz-Sabater, 2021). To match the EM-Earth spatial configuration (a 0.05° offset), the ERA5-Land grids were interpolated, and the combined forcings dataset spanned
120   1950-2023.

The initial (default) SUMMA configuration and parameters used in this study were developed in prior SUMMA and mizuRoute applications projects (e.g., Broman and Wood, 2019; Wood et al., 2021; Wood and Mizukami, 2022), based on expert judgment and review of model parameterizations (i.e., process algorithms) to assess their influence on runoff generation. These choices include model physics selections, soil and aquifer configuration, spatial and temporal resolution, an *a priori* parameter set and
target calibration parameters. The SUMMA model configuration adopted a single HRU per GRU, in which the GRU was the entire lumped area of each catchment. A maximum of 5 layers was specified for snow, and the subsurface included 3 soil layers with a total layer depth of 1.0m, underlain by an aquifer (bucket) with a maximum water holding capacity of 2.5m. For the routing network, the MERIT-Hydro (Yamazaki et al., 2019) topology was chosen.

The SUMMA calibration parameters and physics configuration choices are summarized in Table 1 and Table A1 (appendix),
respectively. The first phase of the calibration process, via a Latin Hypercube Sampling (LHS) of the parameter space and model response, supported sensitivity analysis and refinement of parameter search bounds to focus trial values into likely behavioral areas and/or to avoid model convergence issues (e.g., the lowest theoretically possible values of the vGn_n parameter in SUMMA produces non-physical behavior). The calibration 'trial' parameter selection was designed to control major hydrologic process phenomena -- e.g., infiltration, evapotranspiration, soil storage and transmission, snow accumulation
and melt, hillslope runoff attenuation, aquifer storage and release -- though identifying an efficient number of controlling parameters, versus conducting comprehensive parameter sensitivity assessment and selection optimization.

The 'default' parameter values used here did reflect some calibration effort: they were taken from a prior site-specific CAMELS-based SUMMA streamflow calibration effort that was undertaken in prior project work. That effort (unpublished, led by authors Wood and Mizukami) used the Dynamically Dimensioned Search (DDS) algorithm (Tolson and Shoemaker,
2007) and many of the same parameters, albeit with an earlier version of SUMMA and without mizuRoute routing. The





resulting SUMMA-CAMELS dataset, the DDS calibration workflow and parameter selection was later adopted by a SUMMA sensitivity study (Van Beusekom et al., 2022) and published in associated repositories.

**Table 1. Selected calibration parameters with default values and ranges**

| Parameter Name | Default | Minimum | Maximum | Process Importance |
|---|---|---|---|---|
| k_soil | 7.5e-06 | 1e-07 | 0.0001 | hydraulic conductivity: regulates transmission of water through soil layers |
| theta_sat | 0.55 | 0.2 | 0.7 | soil porosity: influences water storage capacity |
| aquiferBaseflowExp | 2.0 | 1.0 | 4.0 | controls aquifer discharge |
| aquiferBaseflowRate | 0.001 | 0.0001 | 0.1 | controls aquifer discharge |
| qSurfScale | 5.0 | 1.0 | 20.0 | affects partitioning of direct runoff versus infiltration |
| summerLAI | 3.0 | 0.2 | 10.0 | regulates transpiration |
| frozenPrecipMultip | 1.0 | 0.5 | 2.5 | snow undercatch factor, scales winter precipitation |
| routingGammaScale | 18000.0 | 360.0 | 86400.0 | controls GRU combined runoff attenuation and delay |
| routingGammaShape | 2.5 | 1.0 | 5.0 | controls GRU combined runoff attenuation and delay |
| Fcapil | 0.06 | 0.01 | 0.10 | affects refreeze of snowmelt within pack, timing of snowmelt runoff |
| tempCritRain | 273.16 | 270.16 | 276.16 | temperature threshold to discriminate rain from snow |
| heightCanopyTop | 20.0 | 2.0 | 50.0 | impacts turbulent heat fluxes (sensible, latent); influences snow cycle timing and magnitude |
| heightCanopyBottom | 2.0 | 0.000 | 5.0 | not directly calibrated; scaled proportionally to heightCanopyTop |
| windReductionParam | 0.28 | 0.05 | 1.0 | impacts turbulent heat fluxes (sensible, latent) |
| vGn_n | 2.0 | 1.3 | 4.0 | van Genuchten 'n': regulates retention of water in soil layers |

### 2.3 ML-Based parameter estimation approach

We apply and assess the 'large-sample emulator' (LSE) and related parameter estimation techniques introduced in Tang et al. (2024), a companion paper focused on calibrating the hydrological components of the Community Terrestrial Systems Model





(CTSM; Lawrence et al., 2019). For this study, the approach was further tailored to calibrate the combined SUMMA-mizuRoute model (e.g., channel routing was not used with CTSM). As noted in the Introduction, basin-specific calibration approaches can be computationally intensive and result in spatially discontinuous parameter fields, limiting their scalability

and generalizability to large, diverse domains like CONUS. To assess whether the LSE can offer a more effective calibration strategy for large-domain SUMMA modeling, we run several experiments using the ML-based emulation strategy to optimize model parameters, focusing on contrasting two variations: the basin-specific single-site emulator (SSE) and the combined-basin joint LSE, which simultaneously calibrates multiple basins. The calibration period spans six water years, from October 1982 to September 1989, with the first year treated as spin-up and excluded from model evaluation.

### 155 2.3.1 SSE-based calibration

The SSE calibration approach optimizes model parameters for each basin separately. Our approach is based on other single site optimization approaches that use surrogate modeling (e.g., the MO-ASMO method of Gong et al., 2016) to represent the relationship between model parameters and objective function (OF) results. The initial step involves generating a large set (400) of parameter combinations using Latin Hypercube Sampling (LHS) for each basin. These parameter sets are used to run

SUMMA-mizuRoute simulations and their performance is quantified using one or more OFs, which serve as the minimization target for calibration. These initial semi-random sampling and model OF evaluations are first used in selecting the form of the emulator to be used for each basin. Based on performance insights from Tang et al. (2024), two emulators—Gaussian process regression (GPR) and random forests (RF)—are assessed here via a five-fold training and cross-validation procedure on the initial LHS sample. The emulator with better performance is selected and then retrained on the complete initial parameter set,

for each basin separately.

Following this step, the main iterative calibration process begins. For each basin, the trained emulator is used within an optimization algorithm to explore the parameter space, searching for improved parameter sets that minimize the OF. A Genetic Algorithm (GA; Mitchell, 1998) is employed for single-objective calibration, whereas the Non-dominated Sorting Genetic Algorithm II (NSGA-II; Deb et al., 2002) is used for multi-objective calibration. Each iteration involves generating a new suite

(100) of emulator-predicted parameter sets, which are then used to run the SUMMA-mizuRoute model and calculate model OFs. These results are added to the existing previous parameter sets to retrain the emulator for further optimization, leading to the next iteration. This iterative process continues until a specified stopping criterion is met, such as achieving a performance threshold or completing a predetermined number of iterations. In this study, limited iterations (six following the initial LHS iteration '0') combined with a greater number of trials per iteration helped reduce noise while improving calibration efficiency.

The number of trials per iteration and other hyperparameters of the process were selected through sensitivity testing and also informed by the experimental outcomes of Tang et al. (2024).





### 2.3.2 LSE-based calibration

In contrast to the SSE, the LSE calibration approach includes all basins jointly in a single calibration process that estimates optimal parameters in all basins at once. The initial phase is similar to the SSE approach, where a large set (400) of LHS selected parameter sets are generated for each of the 627 basins, and their performance in SUMMA-mizuRoute is evaluated. As in the SSE, the LHS parameter sets are unique for each basin to afford the maximum diversity in parameter trials across the associated model simulations (as in Baker et al., 2021). In contrast to the SSE, however, the emulator must also use static attributes to distinguish between the parameter-performance responses of different basins. In this study, we choose 27 such 'geo-attributes' representing basin-specific geographic and climatic characteristics, such as soil properties, vegetation, and climate indices (Table A2). Including these attributes enables the emulator to estimate model performance (i.e., the OFs) conditioned on both parameter and attribute values, which means the LSE can be used to predict potentially optimal parameter sets for unseen basins where the performance is not known or cannot be measured, enabling its potential use in parameter regionalization, i.e., prediction in ungauged basins.

Figure 2 provides a schematic overview of the ML-based calibration workflow used in this study. We use RF as the emulator form for the LSE due to its speed and performance relative to GPR, which struggles to train on the much larger joint basin parameter trial dataset -- i.e., 400*627 or 250,800 samples for the initial sampling step, growing by 62,700 new trials with each iteration. To start each calibration iteration, the trained RF emulator from the initial step (iteration 0) or prior iteration is used by an optimization algorithm to predict potentially improved parameter sets (100) for each basin individually. In multi-objective optimization, the NSGA-II inherently produces a pareto-set of optimized parameters, whereas for single-objective optimization, we achieve a pareto-set through randomized initializations of the GA. For each new trial in an iteration, static geo-attributes are held constant by restricting their search ranges to the basin-specific values, avoiding the specification of geo-attribute values that do not match those of the study basins. SUMMA is then run with the predicted trial parameters and new OF values are obtained, which are added to the emulator training sample to be used in subsequent iterations.

In collaboration with the effort described in Tang et al. (2024), the development of this calibration approach presented several challenges, which were tackled through extensive testing of different choices in the implementation. For example, a major concern was hyperparameter selection, which required balancing the complexity of the emulator to prevent overfitting while ensuring adequate generalization. Hyperparameters for the RF and GA models were tuned using a combination of grid search and cross-validation. The computational demand of the LSE approach was significant; even using an emulator, it still requires conducting a large number of simulations to generate parameter sets based on optimization algorithms, as well as testing them in a computationally expensive LHM. To address this, the number of iterations was minimized while the number of parameter trials per iteration was increased, which we found improved efficiency without sacrificing accuracy. Additionally, we relied on parallel and high-performance computing (HPC) resources from the National Center for Atmospheric Research (NCAR)





and engineered the HPC-specific workflow using load balancing in the emulator training and parameter optimization phases to reduce the overall computational cost and wall-clock time.

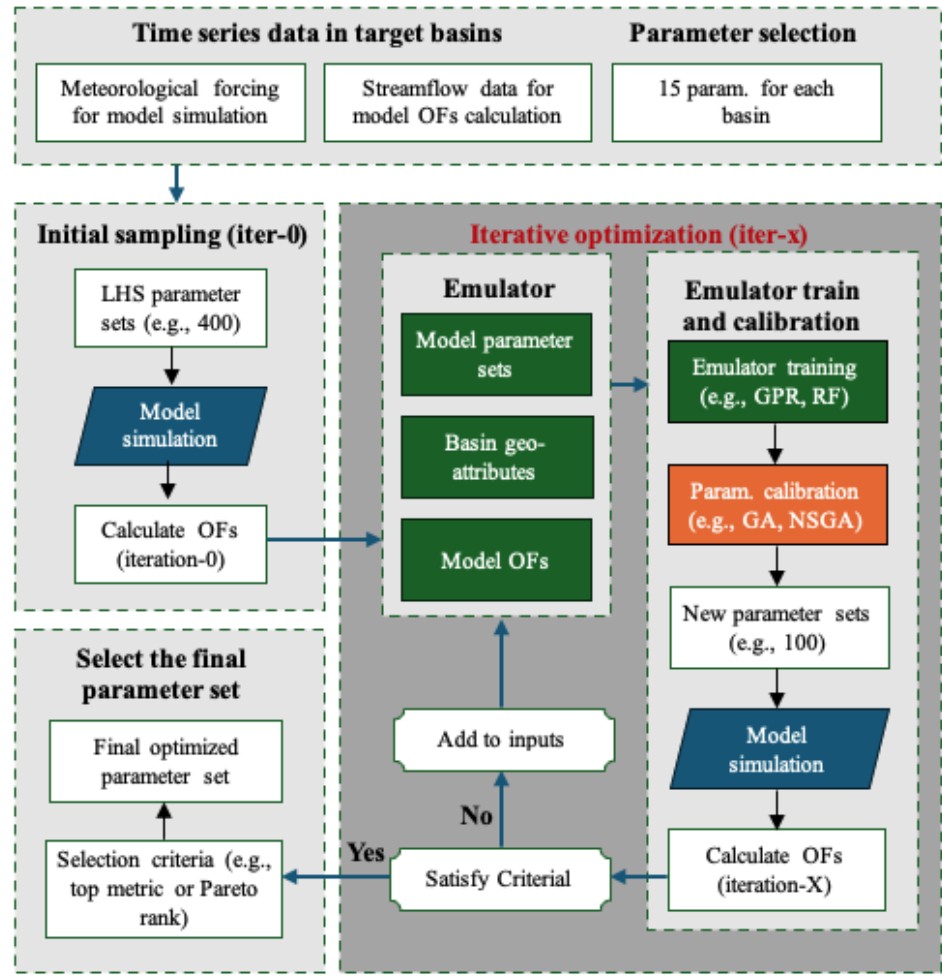


**Figure 2. The flowchart of the parameter optimization in this study, including target parameter selection, initial sampling (iteration-0), iterative optimization (iteration-X) using emulators, and the selection of the final parameter set. For emulator inputs, the basin geo-attributes are only used for LSE.**

**2.4 Experimental design and evaluation approach**

**2.4.1 Experiments**

Our evaluation provides insight into the performance of different aspects of the approach, through several experiments. First, we assess the emulator accuracy -- the agreement between the OFs predicted by the emulator and those simulated by the model. Next, we use a small set of metrics to assess the approach in three ways: (1) with the emulator trained on all of the study basins



('all-basin') for the calibration period; (2) with all-basin training for a temporally separate validation period; and (3) with a
spatial cross-validation, in which the emulator is trained and separately tested on different parts of the study basin dataset.

The first experiment compares the LSE and SSE approach results across all 627 basins during the calibration period. In a
second experiment, temporal validation was performed by selecting the best performing parameter sets from the calibration
period to simulate streamflow for an independent time period (October 2003 to September 2009) to assess the temporal
robustness of the calibrated parameters and their ability to generalize under varying meteorological conditions. This period
has a similar length to the calibration period and is separated by multiple years, without further considerations imposed.

A third experiment (termed LSE_CV) evaluates the LSE's capability for regionalization in unseen basins through using a
spatial cross-validation training and testing approach. The basin dataset was divided through random sampling into five
spatially distinct and (roughly) equally-sized folds. In each iteration of the LSE calibration process, four folds (80% of the
basins) were used for training the emulator and the remaining fold (20% of the basins) was used for testing. Model parameters
(and the emulator-predicted OF values) were predicted by the emulator for the test fold basins based solely on their geo-
attributes. The parameter sets with the best emulator-predicted OFs were then selected for model simulation in the test fold
basins, and OF results for the five test fold simulations were pooled after each iteration for assessment. Note, due to emulator
error, the parameter set selected based on emulator-estimated performance for a test basin was rarely the best performing
parameter set among the iteration trial options for that basin (a point discussed further below). The LSE_CV experiment is
approximately 5 times more expensive than the others, given that 5 emulators must be trained, and after iteration 0, each
iteration involves 5 rather than one model simulation per parameter set.

### 2.4.2 Evaluation metrics and application

The emulator's performance was evaluated using cross-validation techniques and statistical metrics to quantify its ability to
predict the OF values based on parameter sets and catchment geo-attributes. As in Tang et al. (2024), we adopted the
normalized Kling-Gupta Efficiency (NKGE'), a version of the modified Kling-Gupta Efficiency ($KGE$) (Kling et al., 2012;
Beck et al., 2020), as the OF for calibration. NKGE' was chosen to mitigate the influence of outliers, which disproportionately
affect the emulator's performance due to the amplified (unbounded) range of $KGE'$ values from poorly performing basins
and/or trials, and because the joint evaluation requires standardization across diverse basin flow error ranges.

The formulations for KGE' and NKGE' are:

$$KGE' = 1 - \sqrt{(r-1)^2 + (\beta-1)^2 + (\gamma-1)^2} \,, \tag{1}$$

$$NKGE' = KGE'/(2 - KGE') \,, \tag{2}$$

where $r$ is the linear correlation, $\beta$ is the bias ratio, and $\gamma$ is the variability ratio. $KGE'$ ranges from $-\infty$ to 1, while $NKGE'$
normalizes this range to $[-1,1]$, which is necessary to balance the information weight of each basin during training.





For each of the experiments, we take stock of the calibration performance after each calibration iteration (of 100 trials). This
can be done by calculating the evaluation metrics considering each iteration separately, or by calculating them after each
iteration and including prior iterations. The iteration-specific evaluation gives insight into the path of the calibration as it seeks
improved parameters, while the cumulative evaluation shows the overall achievement of the calibration in finding optimal
parameters by the end of each iteration. Most model performance results are shown in terms of the modified Kling-Gupta
Efficiency ($KGE'$) metric (not the optimization metric, which is less familiar to practitioners), and in the form of spatial maps
and cumulative distribution functions (CDFs) of $KGE'$ values from all the basins. Both the LSE and SSE calibrations start with
the same default parameter configurations (iteration-0) to ensure a consistent baseline for comparison. Improvements in $KGE'$
relative to the default model results are analysed in some cases to portray the benefit of both methods.

## 3 Results

### 3.1 Emulator performance

To gauge whether the emulator is successful in learning the model performance metric response to variations in input
parameters (and for the LSE, in geo-attributes), we first evaluate emulator performance by comparing its OF predictions
(NKGE') to the actual SUMMA model OF values across successive calibration iterations. For each iteration, the emulator-
based parameters and OF values are estimated from the simulations of the previous iteration, thus the simulations on which
they are tested are independent (were not seen by the emulator previously). Scatter density plots for iterations 1-6, illustrating
SSE, LSE calibrations and the LSE_CV experiment, are shown in Figs. 3-5, respectively. Figure 3 demonstrates that the SSE
approach struggles to accurately predict the model OF values across iterations. While some improvements are observed (e.g.,
iteration 3), overall performance remains suboptimal, with low to moderate correlations and substantial scatter around the 1:1
line. These results suggest that the SSE approach is limited by a small training sample, resulting in emulator noise and only
weakly capturing the relationship between parameters and OFs, without improving significantly as training progresses. It does
tend to lead to overall model performance improvements as a result of a relatively broad (though inefficient) range of predicted
parameter values in each new iteration (a few of which are often performative).





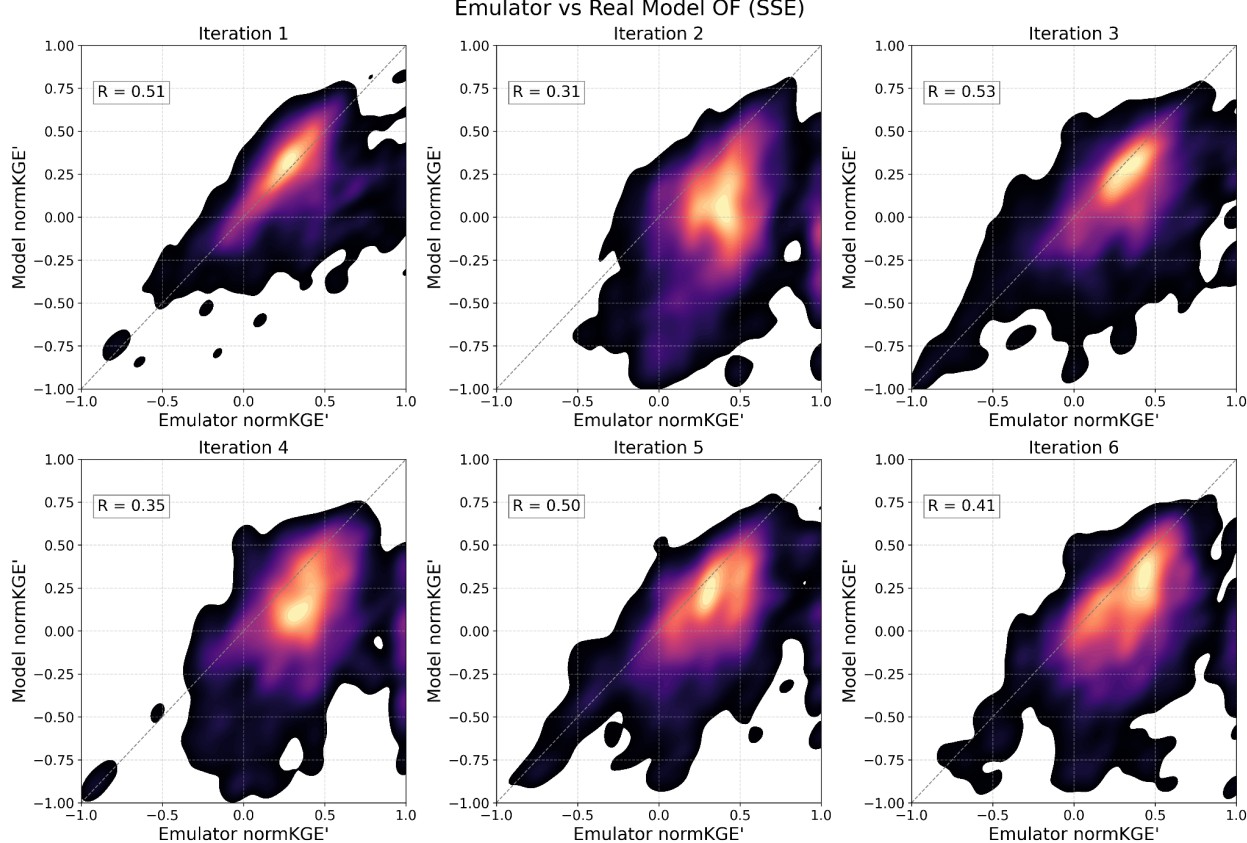

**Figure 3. A scatter density plot of emulator-predicted objective function (OF) values (NKGE') versus real model OF values for the SSE approach across six iterations, with their Pearson correlation coefficient (R) inset.**

Figure 4 demonstrates the LSE approach's progressive and superior ability to accurately predict the model OF values across iterations. Starting with moderate agreement in iteration 1 (R = 0.71), the emulator steadily improves, achieving strong correlations and reduced scatter as training progresses. iterations 4-6, with R fluctuating between 0.90 and 0.94, suggest that the emulator's performance may have reached a limit in the available information. The more progressive upward trend of performance underscores the benefit of the LSE approach relative to the SSE in learning and using the model relationship

between parameters and OFs to more effectively search for parameter sets in the individual basins.





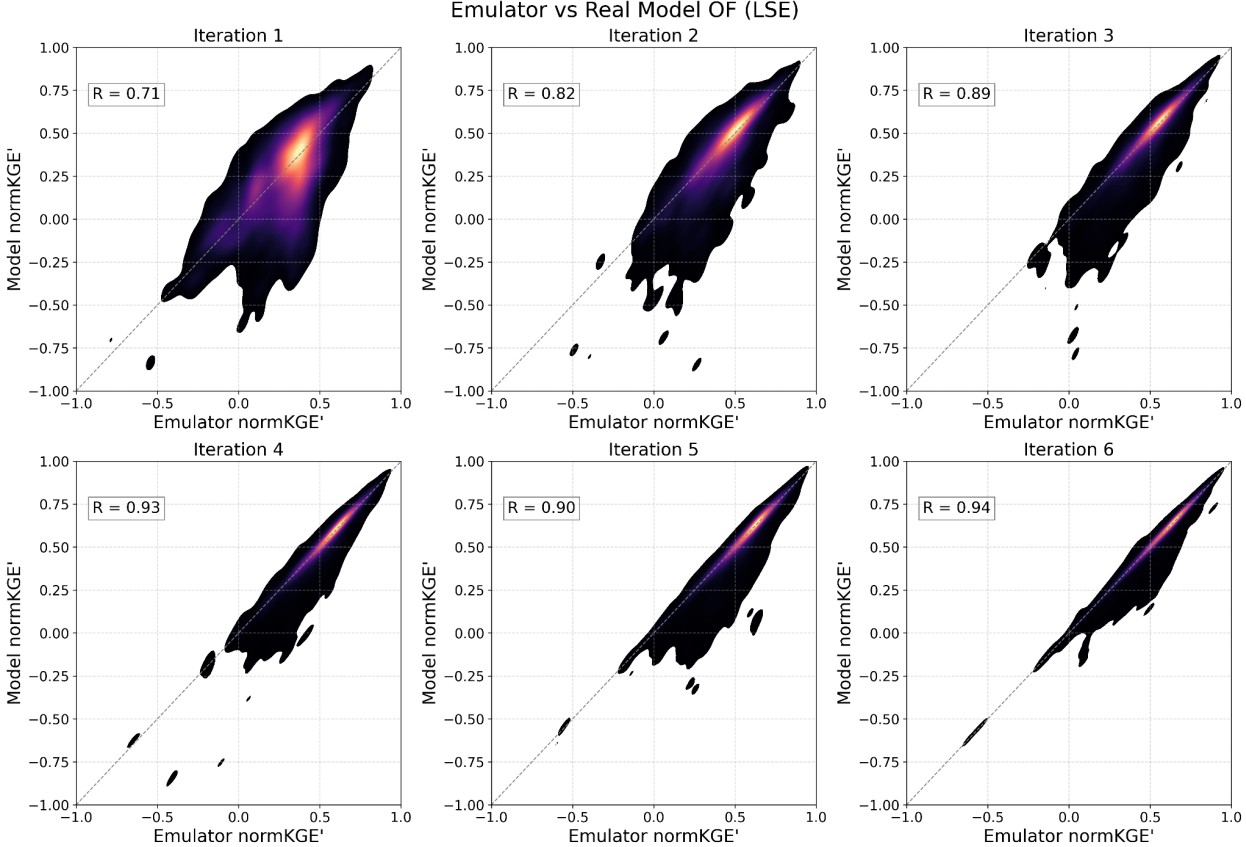

**Figure 4. A scatter density plot between emulator-predicted objective function (OF) values (NKGE') versus real model OF values for the LSE approach across six iterations, with their Pearson correlation coefficient (R) inset.**

Figure 5 evaluates the LSE_CV approach, which represents parameter regionalization in unseen basins through spatial cross-
validation (CV). From the initial iterations (e.g., iteration 1, R = 0.43), the emulator's predictive performance rises more
slowly, with some vacillation, and appears to plateau at much lower skill levels than for the LSE of Figure 4, peaking at R =
0.56 by iteration 5. As expected, the LSE_CV performs worse than the LSE, which sees all basins in calibration, but
nonetheless outperforms the SSE approach (Figure 3). This indicates that the information gained from the large sample of
basins provides additional stability in estimating parameters in an unseen basin over the information gained by the SSE (with
the same number of trials) -- even when the SSE has learned directly (but only) from that basin.





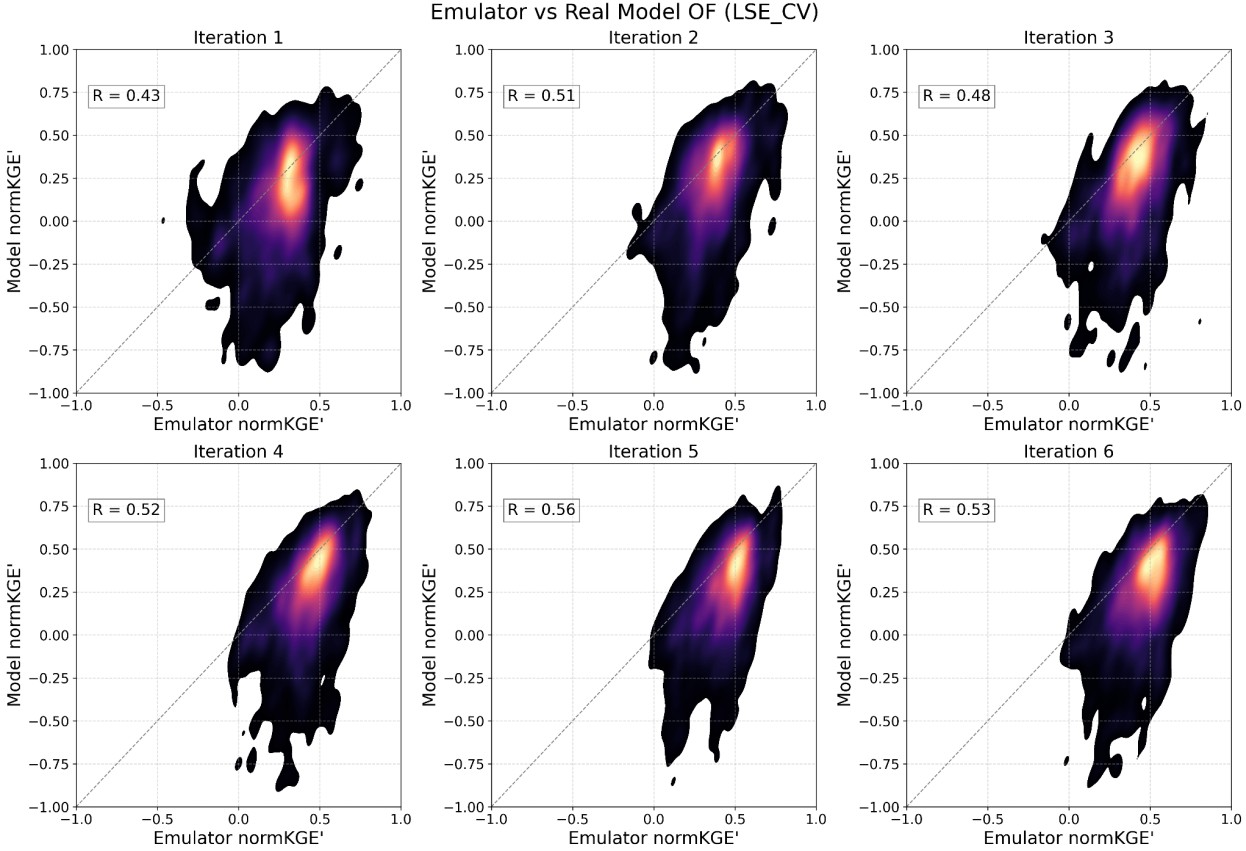

**Figure 5. The scatter density plot between emulator-predicted objective function (OF) values (NKGE') to real model OF values for the LSE_CV approach across six iterations, with their Pearson correlation coefficient (R) inset.**

### 3.2 Calibration performance

The following analyses assess the implications of the apparent emulator strengths and weaknesses for the associated model simulations during calibration. Figure 6 illustrates the CDFs of $KGE'$ for both SSE and LSE calibration approaches across all basins, comparing their evolution from the default parameter set through multiple iterations of calibration. The default configuration provides a baseline with a median $KGE'$ of 0.30, represented by the blue line, and each curve comprises the best 'cumulative' basin model $KGE'$ across all basins after each iteration, including prior iteration results.

The SSE approach (Fig. 6a) shows the SSE calibration results, where the median $KGE'$ improves from the default value of 0.30 to 0.69 by iteration-6. The largest gains are observed during iteration-0 ($KGE'$ of 0.52), after which the improvement rate slows. In Figure 6b, the LSE approach begins with significant improvement in iteration-0, attaining a median $KGE'$ across all basins of 0.52. Over subsequent iterations, the LSE approach gains skill, culminating in iteration-6 with a median $KGE'$ of 0.76. Overall, the LSE approach outperforms SSE across all iterations. By iteration-6, LSE achieves a median $KGE'$ of 0.76



compared to 0.69 for SSE. Both approaches show diminishing returns after early iterations, but the LSE's joint multi-basin calibration is more effective in progressively identifying performative parameter sets through each iteration. Notably, this superior performance is achieved through training a joint model emulator only 6 times, versus training 627*6 = 3,763 emulators to estimate calibration parameters.

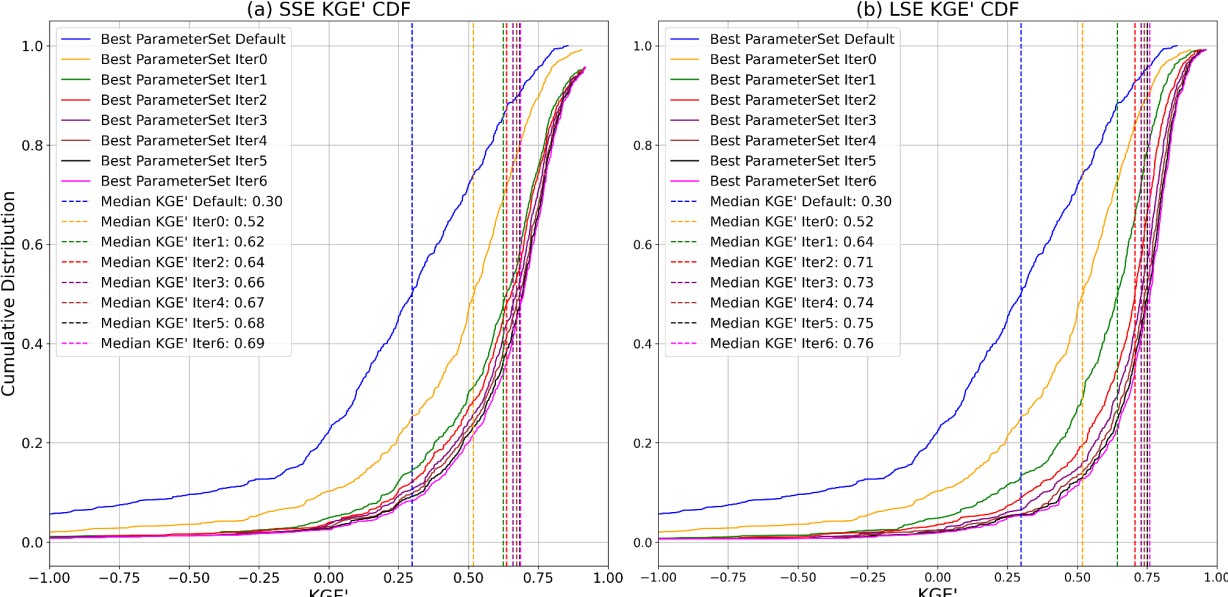

**Figure 6. Comparison of calibration performance: cumulative distribution function (CDF) Comparison of KGE' for (a) SSE and (b) LSE calibration across all basins over six iterations with median KGE' values. The blue line represents CDF and median KGE' based on default parameters of all 627 basins. Both LSE and SSE approaches start with the same iteration-0.**

The geographic distribution of model performance is shown in Fig. 7, which compares individual basin $KGE'$ values for the calibration period across the CONUS domain for the default parameter configuration, the SSE-based calibration, and LSE-based calibration. The default configuration results of Fig. 7a (which reflect a degree of indirect calibration as noted in Sect. 2.2) provide context for this study's calibration improvements. The general pattern of performance, with central US basins showing lower $KGE'$ values than west coast, eastern, and intermountain west bains, is consistent with modeling results shown in numerous other studies based on the CAMELS-US basin dataset, including the first (Newman et al., 2015).

The SSE-based calibration $KGE'$ values (Fig. 7b) show marked improvements in hydrological accuracy relative to the default parameters (Fig. 5a), especially in the eastern United States, where high $KGE'$ values are achieved in many basins. However, the results reveal significant spatial variability, with several western basins showing $KGE'$ values below zero. The LSE (Fig. 7c) demonstrates further improvement across nearly all basins, perhaps most notable in the Appalachian basins of the eastern U.S, and fewer basis with $KGE'$ values below zero (as is also clear from Fig. 6). A timeseries-oriented illustration of the



performance of LSE-based calibration for two basins is provided in Appendix Fig. B1 and Fig. B2. In Figure B1, calibration achieves a high-quality simulation of observed streamflow at daily and seasonal mean levels, while Fig. B2 shows a basin with notable daily and monthly improvements from calibration over the default simulation, but errors remaining in the seasonality of simulated streamflow.

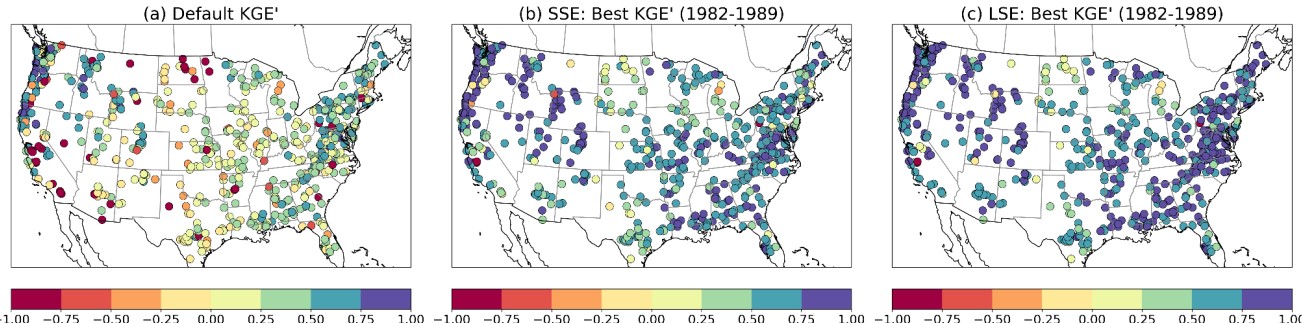

**Figure 7. Comparison of KGE' values for (a) default configuration, (b) SSE and (c) LSE calibration across the CONUS**
**(1982-1989).**

Although such CONUS-wide summaries of performance are useful, the contrast between SSE and LSE calibration performance can be stark when reviewed at the level of individual basins. Figure 8 shows an example of the parameter sampling trajectories of the SSE and LSE calibrations, as assessed using two model diagnostic performance metrics that were not used as the calibration objective: the mean absolute daily streamflow error, and a seasonality metric defined as the maximum long-
term mean monthly absolute streamflow error -- i.e., the largest error in long-term mean monthly flow. The LHS-generated 'Iter-0' parameters provide the foundation for searching the parameter space, which the SSE and LSE duly improve upon, each recommending parameters that lead to superior model performance in Iteration 1. In subsequent iterations, however, the SSE parameter recommendations regress and vacillate in model performance, while the LSE parameter recommendations tend to advance. For the SSE, the successive iterations of the parameter search are have broader spread and are more scattershot, while
the LSE search tends to yield steady progress in model performance. This behavior certainly varies by location, but the general character of each approach shown in Fig. 8 is corroborated by the rates of improvement shown in Fig. 6.





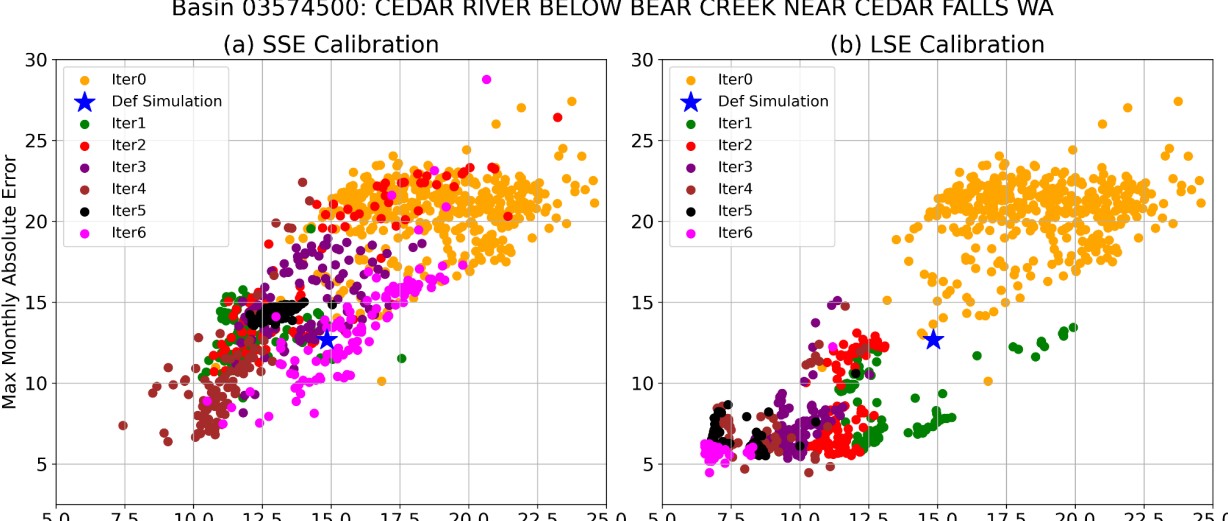

**Figure 8. Illustration of the (a) SSE and (b) LSE parameter calibration progress across successive iterations, as measured by two metrics not used in calibration. Better model performance for both metrics is found in the bottom left corner of each plot. Iteration-0 contains 400 parameter sets, and subsequent iterations contain 100. The default simulation is from a previous SUMMA application after individual basin optimization with the DDS algorithm.**

Incidentally, Figure 8 also illustrates that despite recent critiques against calibrating hydrology models to integrated streamflow metrics such as the Nash Sutcliffe Efficiency (NSE) and KGE (e.g., Brunner et al., 2021; Knoben et al., 2019), such metrics can be effective in jointly optimizing hydrology model performance across multiple dimensions, hence their long-standing popularity in practice.

### 3.3 Temporal validation of the SSE and LSE approaches

Temporal validation (during the independent 2003–2009 period) of the SSE and LSE approaches shows that, as expected, calibration parameter performance for both falls relative to the calibration period. Figure 9 shows the $KGE'$ CDF curves with the median of the distribution of each basin's best performance (including all prior iterations) reaching 0.65 and 0.69 or the SSE and LSE, respectively, by the 6th iteration (best scores include the best of all prior iterations). The lower reduction in validation scores ('shrinkage') for the SSE than for the LSE may or may not be notable (i.e., it may be a study-specific result); if significant, it suggests that the best selected SSE parameters, even with lower overall performance in both calibration and temporal validation, may be slightly more robust to meteorological variability than those from the LSE.



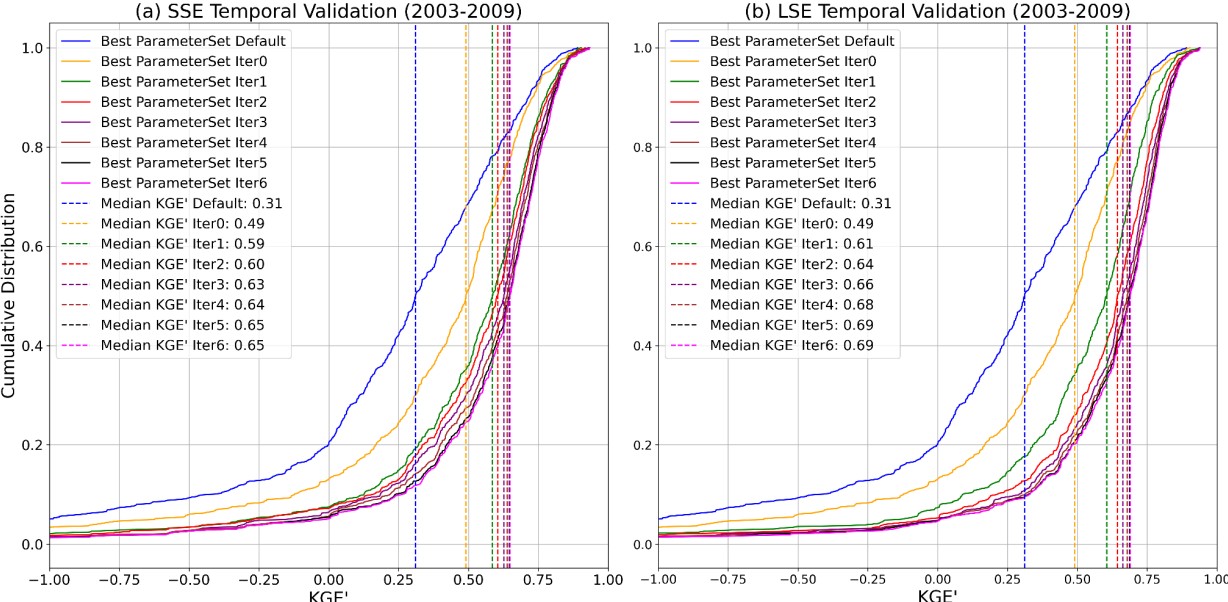

**Figure 9. Comparison of temporal validation performance: CDFs of the best KGE' across all basins over six iterations for the (a) SSE Calibration and (b) LSE calibration. The blue line represents the CDF and median KGE' based on default parameters over all basins.**

The associated maps in Figure 10 show the geographic distribution of the temporal validation median $KGE'$ scores for the SSE and LSE approaches, and their change in value relative to their respective calibration scores. The pattern of values for the validation scores (parts a and c) are broadly similar, which is notable given that the LSE represents their joint calibration in contrast to the individual attention that each basin receives in the SSE. Higher $KGE'$ values are observed predominantly in the mountainous portions of the western US, and in the midwest and eastern US. Lower $KGE'$ values are more prevalent in the southwestern US and northern plains region.

In general, areas that calibrated well under either method (Fig. 7) tended to hold up well in temporal validation. Plot parts (b, d) show the difference between the validation and best-calibrated $KGE'$, and regions with smaller differences indicate that the parameter sets obtained during calibration generalized well in time. The LSE-calibrated parameters led to slightly greater loss in skill in validation than did the SSE-calibrated parameters, and this effect was pronounced in the more challenging calibration regions noted above. Although the cause of this effect is unclear, a likely culprit is overtraining -- i.e., that the LSE harnesses more sequence-specific information than the SSE to gain a stronger calibration and validation performance. That said, if the objective of a modeling application is to calibrate a LHM for use over a large number of measured catchments, this analysis nonetheless suggests that the LSE would provide both efficiency and skill improvements over the traditional site-specific calibration.




**Figure 10. Temporal validation of KGE' for SSE and LSE calibration across the CONUS.**

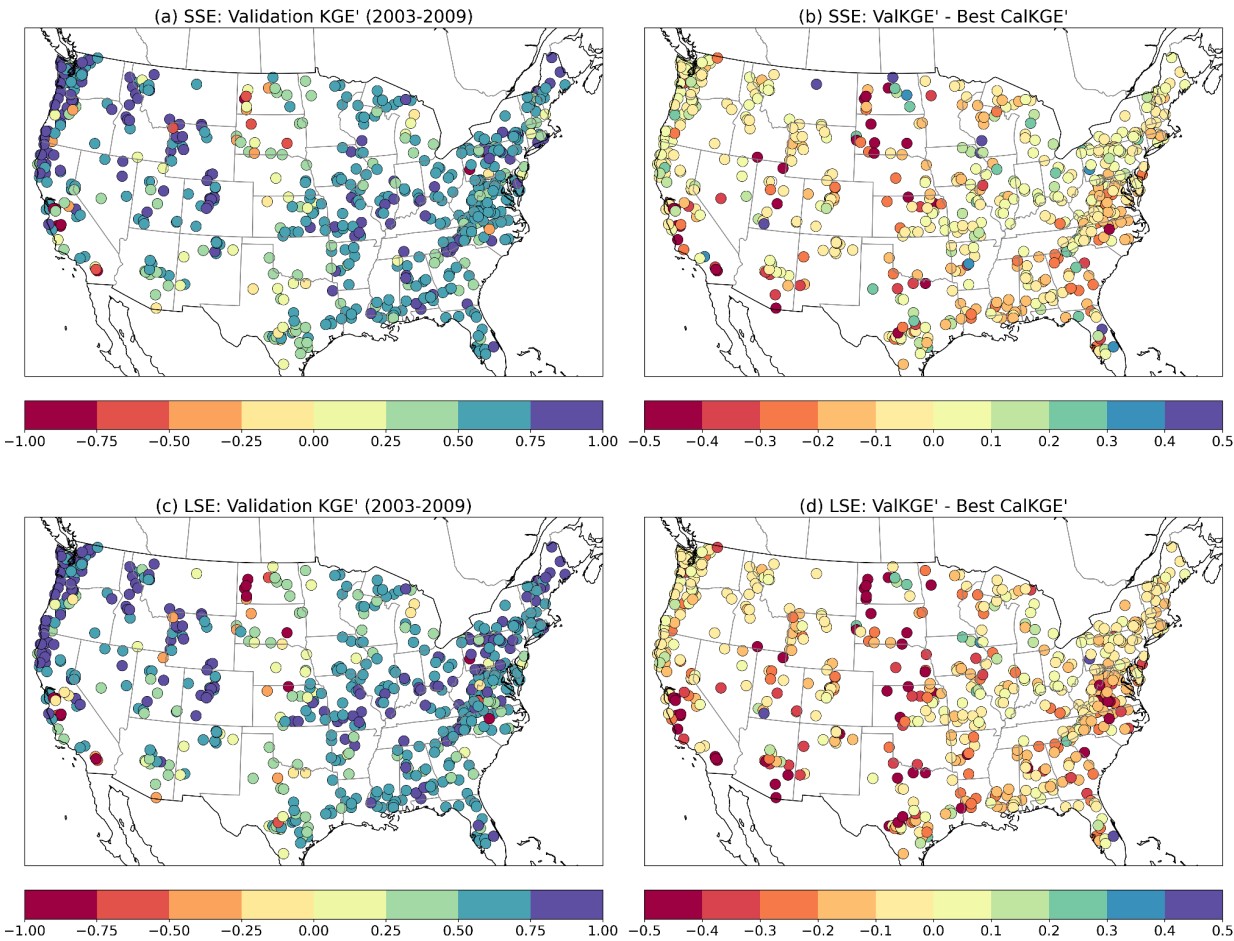

## 3.4 Spatial cross-validation

Figures 11 and 12 present results of the LSE_CV experiment, which tests SUMMA simulations using the parameter sets in each iteration (cumulative with prior iterations) that had the best emulator-predicted $KGE'$ values in each basin. Each calibration iteration produces 100 recommended new parameter sets; thus, there is a need to decide *a priori* which set to select for testing in the unseen basins (as noted in Sect. 2.4.1). Because the emulator has some skill in estimating model performance given different parameter sets, we use its performance estimate as a basis for the selection. We briefly experimented with

alternative selection strategies, none of which were superior, and also evaluated whether transferring a small ensemble (top 5-20 parameter sets based on emulator ranking) leads to better mean performance (it does, but ensemble modeling is not the focus of this effort). We compare LSE_CV results to those from LSE calibration parameter sets selected in the same way




(based on the highest emulator-predicted $KGE'$ values), which leads to slightly lower performance than assessing the best actual model $KGE'$ values, as shown in Fig. 6a).

In Figure 11, the contrast between the LSE_CV and LSE_all (from calibration) when using emulator-ranked parameter sets is striking. In both cases, median $KGE'$ values generally rise over six iterations, indicating that the approaches can find improved parameters as they are run repetitively. Not surprisingly, the LSE_CV skill falls relative to the LSE all, and plateaus quickly. The LSE_all calibration achieves higher median emulator-predicted $KGE'$ values compared to LSE_CV in every iteration, with a final median value of 0.73 for LSE_all compared to 0.43 for LSE_CV by iteration 6. This discrepancy was larger than

expected and may arise for reasons which are explored further in the Discussion section.

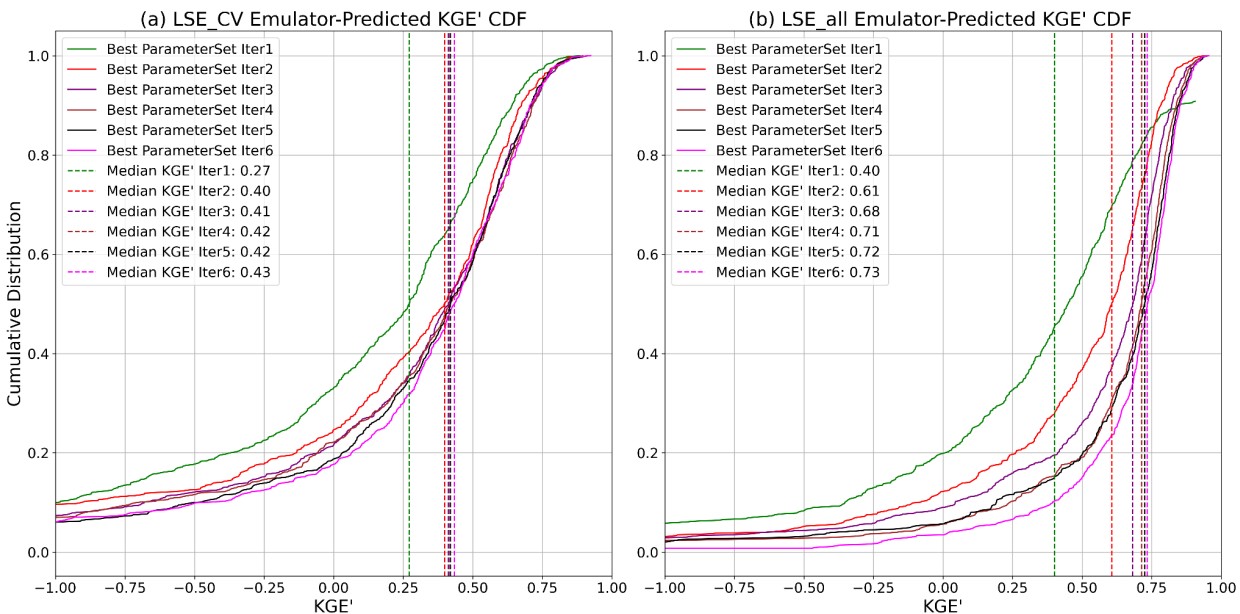

**Figure 11. Comparison of calibration performance: cumulative distribution function (CDF) Comparison of KGE' for (a) LSE_CV across all folds and (b) LSE_all across all basins over six iterations with median KGE' values.**

Noteworthy context is provided in Appendix Fig. B3, which shows that the best model performance achieved in test basins

using LSE_CV parameter sets (recall that 100 parameters sets are estimated in each iteration, but only one is tested in Fig. 11a). Achieving a median $KGE'$ value of 0.72 after six iterations, the best performing results suggest that the LSE is not incapable of finding competitive parameter sets in unseen basins; rather, the challenge is knowing in advance which of the estimated parameter sets are best to use in parameter transfer. For applications in which an ensemble of parameter sets is useful, this finding is useful, especially if the fitness of an ensemble of emulator-predicted parameter sets could be judged

*post-facto* through additional relevant criteria (such as performance at indirectly related or downstream gages), aiding the regionalization task.



Figure 12 shows the geographic distribution of performance results for the LSE_CV, the LSE in calibration (which uses emulator-ranked best parameter sets, versus best model outcomes), and the LSE_CV difference from the default model performance (parts a-c, respectively). The calibrated LSE-based $KGE'$ values are significantly high across most basins,
demonstrating that substantial model performance may be achieved by directly calibrating parameters across all basins, benefiting from the full training dataset. The more uniform distribution of higher $KGE'$ values across different regions, especially in the central and eastern U.S., highlights the LSE's ability to enhance accuracy over diverse hydroclimatic conditions. The LSE_CV approach, illustrated in Fig. 11a, shows improvement over the default model parameters for a majority of basins, with approximately 62% of basins achieving better values than the default configuration. However, there
are multiple basins where LSE_CV underperforms compared to default parameters, particularly in complex regions.

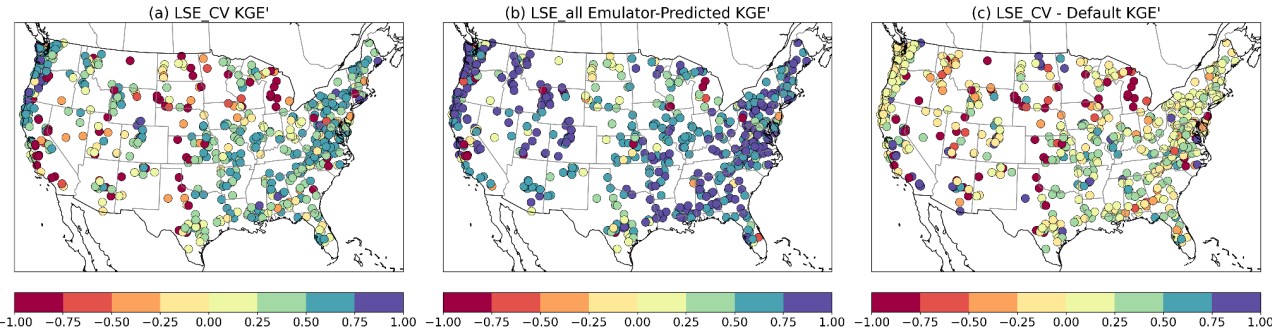

**Figure 12. Comparison of performance (median sample KGE') between the LSE_CV, which uses spatial cross-validation for regionalizing parameters to unseen basins, and the LSE_all calibration, which uses information from all basins. The LSE_CV performance difference from the default parameter performance is also shown.**

## 5 Discussion and Conclusions

This study investigates whether the challenge of calibrating large-domain implementations of complex, expensive process-based land/hydrology models can be tackled through similar strategies to those now being advanced in the AI contexts (i.e., ML, DL and differentiable modeling). Such studies have shown that jointly training the DL or simple conceptual models (made differentiablee) over large samples of catchments is not only viable, but is recommended over individually calibrating such models on single basins. The reverse had generally been found true for complex LHMs, but the recent emergence of ML model
emulation strategies for complex models has provided an avenue for reassessing this consensus. In collaboration with a companion effort described in Tang et al. (2024), which focused on the CTSM land model, we develop and assess a large-sample emulator (LSE) based strategy for calibrating the SUMMA-mizuRoute modeling approach across CONUS watersheds.

Our findings are, in short, promising. They suggest that a large-sample model response emulation approach has potential to become a preferred option for calibrating complex PB models over large domains as it has for ML and other AI-era modeling
approaches. The generally higher performance achieved by the LSE relative to the SSE indicates that large-sample calibration





can more effectively learn the model response surface to parameters, even for complex models, than is possible using local information alone. As noted by Kratzert et al. (2019) and others, the inclusion of static catchment attributes allows large-sample approaches to localize parameter influence and account for hydroclimatic variability across basins, which in turn leads to more efficient joint calibration and better overall model performance. Our SSE, while effective for some individual basin
calibrations, struggled to reach the accuracy of the LSE when applied across diverse conditions, and pursued less efficient parameter search trajectories. In practice, the scalability of a strategy that jointly trains a single, low-cost model emulator for model calibration to yield usable parameter estimates for hundreds (or more) catchments at once is arguably attractive, given the alternative of individually calibrating at least a fraction of those catchments only as the first step toward training a separate parameter transfer scheme.

The results of the LSE calibration (median $KGE' = 0.76$) and validation (0.69) in this study are competitive with published modeling studies using all or parts of the CAMELS catchment collection, though comparisons are inexact due to differences in factors such as basin selection, validation periods, and optimization objectives. For example, Feng et al. (2022) reported median temporal validation NSEs ranging from 0.62 to 0.75 for jointly calibrated DL and differentiable learning models, while Newman et al. (2015; 2017) achieved sample median Nash Sutcliffe Efficiency (NSE) scores around 0.74 for calibration and
0.60–0.70 during temporal validation, using much simpler conceptual models (Sacramento and Snow-17), all individually trained.

Yet in other regards, such as advancing capabilities for prediction in ungauged basins, it is also clear from these experiments that further understanding and improvements are needed. The median $KGE'$ score of 0.43 achieved in spatial cross validation is likely inadequate for use in many regionalization applications, though we believe it is on par with what is currently
achievable for complex PB models using site-specific basin calibration followed by similarity-based regionalization. For instance, the performance is near the ungauged basin evaluation over CONUS reported in Song et al. (2024) for the US National Water Model 3.0, at KGE = 0.467. It moderately lags a new differentiable physics-informed ML model (δHBV2.0δUH) at 0.553 in the same study, and considerably lags results from pure DL approaches -- e.g., the impressive median NSE of 0.69 achieved by the PUB LSTM of Kratzert et al. (2019). Such studies are not controlled comparisons with this one or each other,
but nonetheless provide useful context.

The lower performance of the LSE_CV relative to the LSE in spatial cross-validation validation may result from a combination of factors, including unexplained variability in the hydroclimatic settings and model response, and some overtraining to sample characteristics, which include meteorological input errors. This can also be attributed to (1) the reduced number of training samples in LSE_CV compared to LSE_all (20% fewer basins), and (2) the inherent difficulty of regionalizing parameters for
ungauged locations, a well-documented challenge in hydrological modeling (Patil and Stieglitz, 2015). Overall, the sample size used in this study (627) may be inadequate for high-quality regionalization. Nonetheless, we are optimistic that with further exploration and development, the regionalization performance of calibration based on an LSE approach will improve.



We have not yet explored potential refinements such as feature engineering and selection of static geo-attributes to enhance transferability. Here we simply adopted those used in Tang et al. (2024), and these contained inconsistencies (e.g., meteorological attributes were not based on the model forcing dataset climatology). Using more catchments in training with better screening for representativeness is likely to strengthen the regionalization, especially as some basins were later found to have erroneous streamflow observations. Supporting work in the study (not shown here) indicated that some parts of the US improve when restricting training to a similarity-based watershed selection, while others fare better when trained on the full sample – thus a blend of similarity-based and full-domain emulation may prove superior. The selection strategy for predicted parameter sets to transfer and various hyperparameter choices also warrant further investigation. This effort along with Tang et al. (2024) are initial attempts at implementing such a ML-based joint calibration strategy, and raise as many questions as they answer.

Overall, we hope that these findings will update conventional wisdom about the ability of complex process-based LHMs to compete with simpler conceptual models in performance, given that our temporal validation across hundreds of basins is on par with that of other published CAMELS-based conceptual modeling studies. Perhaps more importantly, we show that the power of large-sample model training underpinning recent advances in ML hydrology is extensible to complex process-based hydrology models as well. We believe the work takes an important step toward addressing the longstanding challenge of applying such models of prediction in ungauged basins. With national water agencies and global modeling initiatives for land/hydrology and climate analysis and prediction continuing to seek unique multivariate insights from complex process-based land/hydrology modeling approaches, we encourage further exploration of possibilities in this direction.

## Appendix A. SUMMA configuration

**Table A1. SUMMA default model decisions (physics configuration) for this study.**

| Model decision | Chosen option | Model decision description |
|---|---|---|
| soilCatTbl | STAS | soil-category dataset |
| vegeParTbl | MODIFIED_IGBP_MODIS_NOAH | vegetation category dataset |
| soilStress | NoahType | function for soil moisture control on stomatal resistance |
| stomResist | BallBerry | function for stomatal resistance |



| num_method | itertive | choice of numerical method |
|---|---|---|
| fDerivMeth | analytic | method used to calculate flux derivatives |
| LAI_method | specified | method used to determine LAI and SAI |
| f_Richards | mixdform | form of Richard's equation |
| groundwatr | bigBuckt | choice of groundwater parameterization |
| hc_profile | constant | choice of hydraulic conductivity profile |
| bcUpprTdyn | nrg_flux | type of upper boundary condition for thermodynamics |
| bcLowrTdyn | zeroFlux | type of lower boundary condition for thermodynamics |
| bcUpprSoiH | liq_flux | type of upper boundary condition for soil hydrology |
| bcLowrSoiH | drainage | type of lower boundary condition for soil hydrology |
| veg_traits | Raupach_BLM1994 | parameterization for vegetation roughness length and displacement height |
| canopyEmis | difTrans | parameterization for canopy emissivity |
| snowIncept | lightSnow | parameterization for snow interception |
| windPrfile | logBelowCanopy | wind profile through the canopy |
| astability | louisinv | stability function |
| canopySrad | BeersLaw | canopy shortwave radiation method |
| alb_method | conDecay | albedo representation |
| compaction | anderson | compaction routine |
| snowLayers | CLM_2010 | method to combine and sub-divide snow layers |





| thCondSnow | jrdn1991 | thermal conductivity representation for snow |
| thCondSoil | funcSoilWet | thermal conductivity representation for soil |
| spatial_gw | localColumn | method for the spatial representation of groundwater |
| subRouting | timeDlay | method for sub-grid routing |

**Table A2. Geo-attributes used in the large-sample emulator (LSE) training**

| Attribute Name | Relevance | Description | Unit |
| --- | --- | --- | --- |
| mean_elev | Topography | catchment mean elevation | m above sea level |
| mean_slope | Topography | catchment mean slope | m/km |
| area_gauges2 | Topography | catchment area (GAC) | km$^2$ |
| p_mean | Climate | mean daily precipitation | mm/day |
| pet_mean | Climate | mean daily PET (estimated) | mm/day |
| aridity | Climate | aridity (PET/P ratio) | - |
| p_seasonality | Climate | seasonality and timing | - |
| frac_snow | Climate | fraction of precipitation as snow | - |
| high_prec_freq | Climate | frequency of high precipitation | days/year |
| high_prec_dur | Climate | average duration of high precipitation | days |
| low_prec_freq | Climate | average frequency of low precipitation | days/year |
| low_prec_dur | Climate | average duration of low precipitation | days |
| frac_forest | Landcover | forest fraction | - |
| lai_max | Landcover | maximum monthly LAI | - |
| lai_diff | Landcover | difference between max and min LAI | - |



| dom_land_cover | Landcover | dominant land cover type | - |
| dom_land_cover_frac | Landcover | fraction of the catchment area of dominant land cover | - |
| soil_depth_pelletier | Soil | depth to bedrock | m |
| soil_depth_statsgo | Soil | soil depth (maximum) | m |
| soil_porosity | Soil | volumetric porosity | - |
| soil_conductivity | Soil | saturated hydraulic conductivity | cm/h |
| max_water_content | Soil | maximum water content | m |
| sand_frac | Soil | sand fraction | % |
| silt_frac | Soil | silt fraction | % |
| clay_frac | Soil | clay fraction | % |
| carbonate_rocks_frac | Geology | fraction of the catchment with carbonate rocks | - |
| geol_permeability | Geology | subsurface permeability | $m^2$ |

485





**Appendix B**

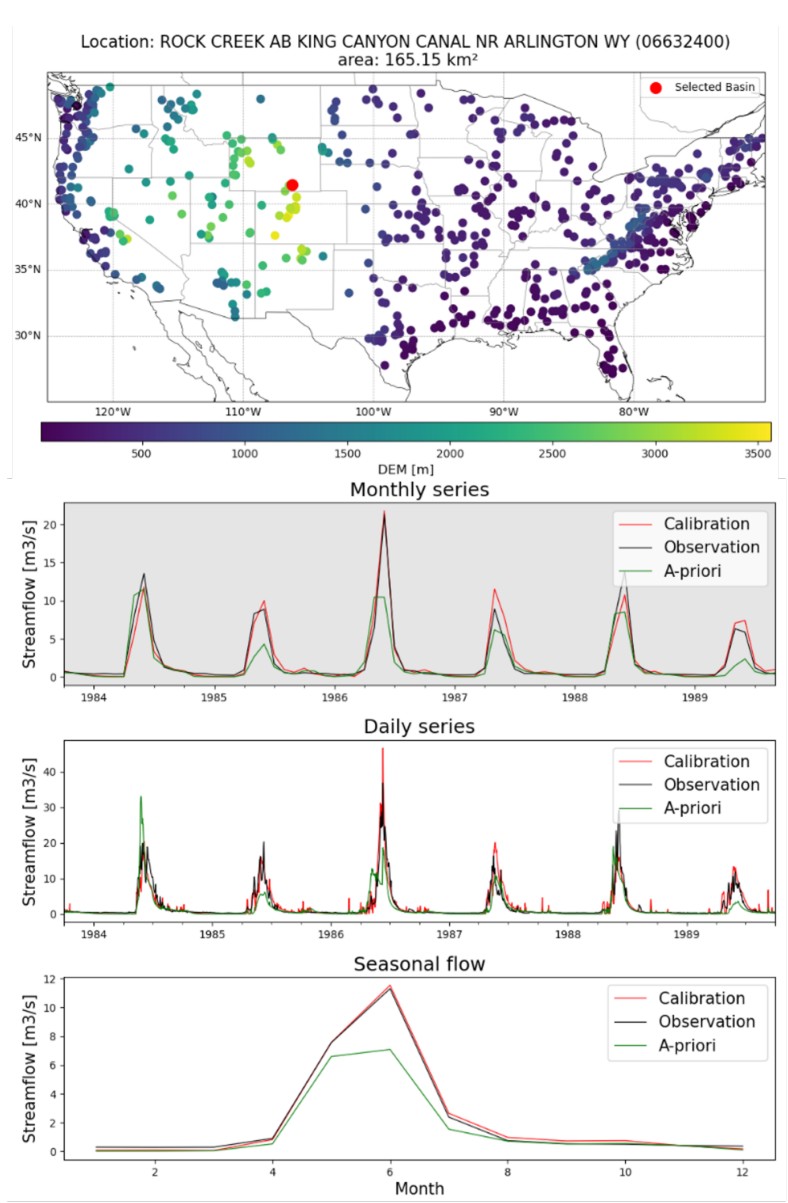

**Figure B1. An example basin illustrating how simulated streamflow using LSE calibrated parameter values aligns significantly better with observations compared to simulations using a-priori (i.e., default) parameters.**





490

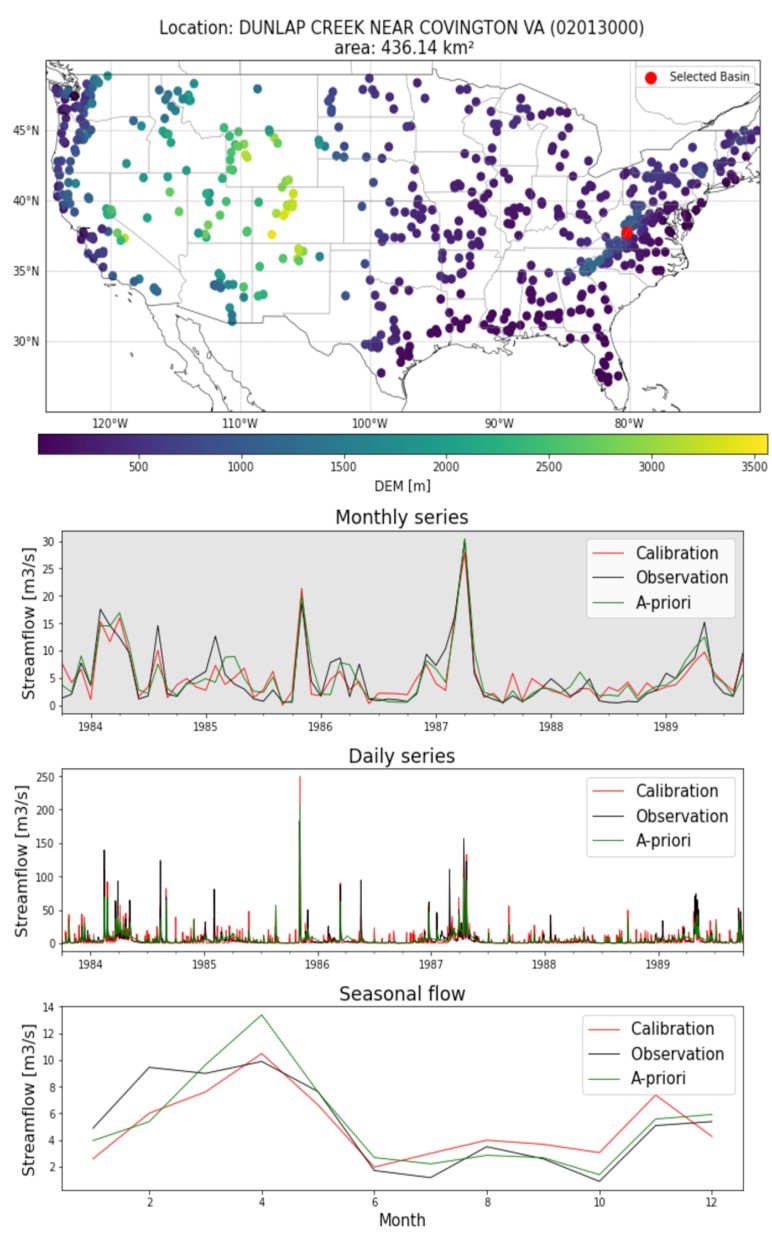

**Figure B2. Same with Fig. B1 but showing an example basin where the LSE calibration shows lesser improvement regarding the seasonal streamflow compared to a-priori (i.e., default) parameters, while for the daily and monthly series, the improvement is still notable.**



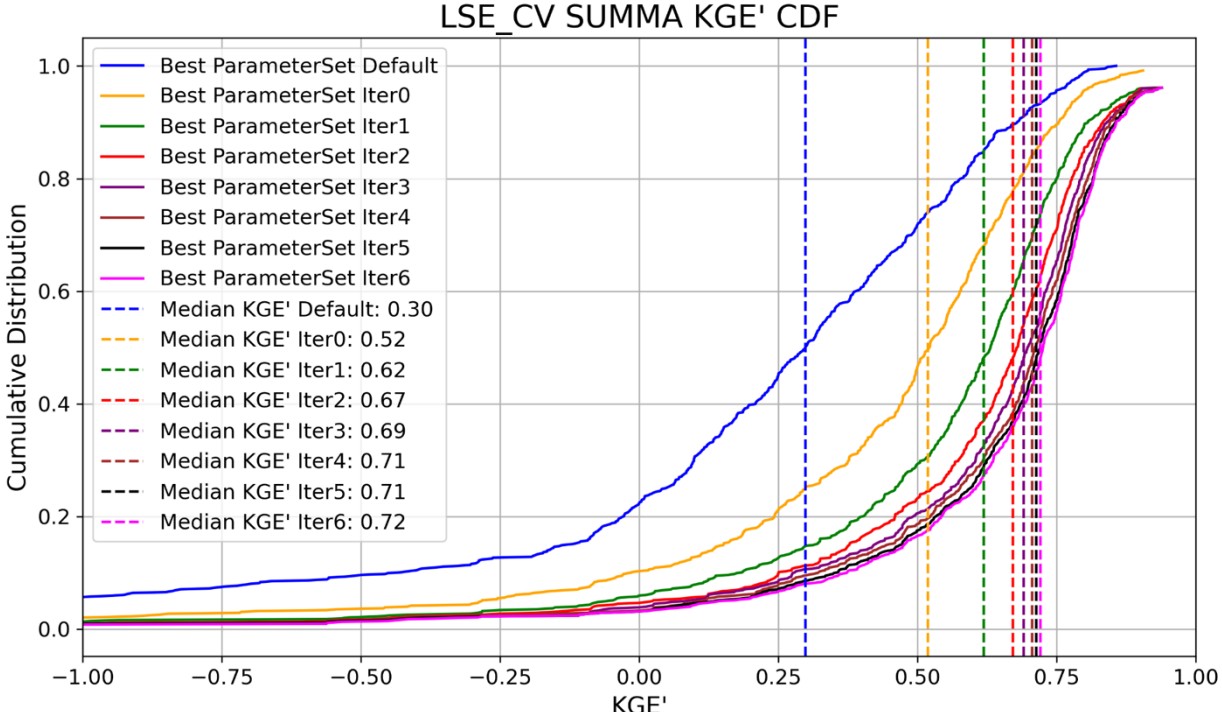

**Figure B3. Comparison of the best post-facto model KGE' CDFs for LSE_CV over six iterations, and including sample median KGE' values.**

## Code and data availability

The SUMMA model is available at https://github.com/CH-Earth/summa/ and the mizuRoute model is available from https://github.com/ESCOMP/mizuRoute. The original CAMELS dataset is available at https://gdex.ucar.edu/dataset/camels.html. EM-Earth is available at https://doi.org/10.20383/102.0547. ERA5-Land data is available at https://doi.org/10.24381/cds.e2161bac. The LSE-based optimization codes and associated datasets will be shared on an open-access platform after manuscript publication.

## Acknowledgments

We acknowledge valuable guidance, support and feedback from Chanel Mueller (USACE) and Chris Frans (Reclamation), which helped to motivate and steer this effort toward US water agency relevance and application.



**Financial support**

This study is supported by the research grants to NCAR from the NASA Subseasonal-to-Seasonal Hydrometeorological Prediction Program (Award #80NSSC23K0502), the US Army Corps of Engineers Climate Preparedness and Resilience Program (Agreement #HQUSACE17IIS), the US Bureau of Reclamation Research and Development Office (Award # R24AC00121), and the NOAA Climate Observations and Modeling Program (Award #NA23OAR4310448). We acknowledge high-performance computing support provided by NCAR's Computational and Information Systems Laboratory, sponsored by the US National Science Foundation.

**Author contributions**

MF and AW wrote the manuscript, and AW, MF, and GT designed the study and contributing analyses. GT developed the core calibration codes, workflows, and forcing dataset, with strategy and design guidance from AW (see Tang et al., 2024), and assisted MF in adapting these workflows and techniques to SUMMA. NM also helped to incorporate channel routing with mizuRoute. MF ran all simulations, analyzed and refined the approaches, and prepared all of the figures with assistance from GT. GT and NM provided final manuscript edits and feedback. AW leads the research projects sponsoring this work.

**Competing interests**

The authors declare that they have no conflict of interest.

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
