# Peer review of "Calibrating a large-domain land/hydrology process model in the age of"

_EGUsphere, 2025_

## Author Response (AR1)

**Referee #1 comment for SUMMA HESS preprint**

*We thank Refree #1 for constructive comments. Here we provide our responses to comments from referee #1. The original review comments are in* black, *and our responses in red, italic font.* Blue *text was incorporated into the revised paper.*

Dear Editor,

In the manuscript 'Calibrating a large-domain land/hydrology process model in the age of AI: the SUMMA CAMELS experiments' the authors present a novel hydrological model calibration method. Using a machine learning model, the authors map directly from the calibration parameters, and catchment attributes for the generalized calibration experiment, to the model performance. Subsequently, increasingly better calibration parameters are iteratively selected by using a genetic algorithm in tandem with the machine learning model, updating the machine learning model when new results are in. The manuscript is well written, thorough, and relevant, although some parts of the manuscript remain vague and could be improved. Therefore, I would recommend minor revisions for this manuscript. Below is a more expansive description of my main arguments, as well as a list of line-by-line comments.

Vague manuscript sections

Although the manuscript is well written, the novel calibration approach introduced in the manuscript remains unclear and in the background throughout the manuscript (except for the methods). The title, abstract and introduction could be improved by clearly stating what this study has done, instead of which model or which dataset was used. The same holds true for the discussion and conclusions, where more focus should be on the specific contribution of this study's calibration approach, instead of generalization statements already made by various other studies. See the below line-by-line comments for more details.

Specific comments

Title: The title is not very descriptive and does not capture the study well. There are many different studies that calibrate large-domain hydrological models using AI. I would suggest revising the title.
*Response: Thank you for this suggestion. We have revised the title to explicitly highlight the* **emulator**-*based calibration method: "Calibrating a large-domain land/hydrology process model in the age of AI: the SUMMA-CAMELS* emulator *experiments". Overall, we like the title because it captures both the general context of the study, ie that opportunities for calibration are changing in this new era of AI methods (where AI has become a common umbrella term for ML, DL, generative AI and other techniques), and also that the study focuses on a particular set of experiments conducted with a recognizable model and dataset.*

*We somewhat disagree that there are many different studies calibrating large domain **process-based/complex** models using AI ... versus the many studies calibrating ML or simple conceptual models. There are actually fairly few studies (as we explain in the intro) using a fully dynamical model emulation approach for complex models, and none that have tried this large-sample response emulation approach to date, except for the companion paper that we reference, Tang et al (2025).*

Line 12: "a machine learning (ML) based calibration strategy": What are the novel aspects of this strategy? This provides little information the study.

*Response: We revised the abstract to emphasize that the emulator joint training and potential for regionalization is the novelty of this work. The abstract is necessarily concise and further information is provided in the manuscript.*

Lines 15-18: "the large-sample emulator (LSE) approach" / "a single-site emulator (SSE)" these terms are very unclear as they have not been properly introduced.

*Response: We revised the abstract to clarify LSE and SSE:* "This study introduces a new scalable calibration framework that jointly trains a machine learning emulator for model responses across a large-sample collection of watersheds while leveraging sequential optimization to iteratively refine hydrological model parameters. We evaluate this strategy through a series of experiments using the Structure for Unifying Multiple Modeling Alternatives (SUMMA) hydrological modeling framework coupled with the mizuRoute channel routing model for streamflow simulation. This 'large-sample emulator' (LSE) approach integrates static catchment attributes, model parameters, and performance metrics, and yields a powerful new strategy for large-domain PB model parameter regionalization to unseen watersheds. The LSE approach is compared to using a more traditional individual basin calibration approach, in this case using a single-site emulator (SSE), trained separately for each basin."

Line 64: "physics-based PB": double

*Response: Corrected.*

Lines 71: "model emulation": This study does not actually emulate the model, but the model performance. This distinct difference should be made more clear, especially as this is contrary to most of the other studies discussed in the introduction.

*Response: We added a paragraph in introduction to explain how different studies use emulators:* "Generally, emulator strategies have evolved along two primary lines: (i) emulating model performance by directly relating model parameters to one or more performance objective functions, without explicitly modeling the dynamic behavior of the system (Gong et al., 2016; Herrera et al., 2022; Maier et al., 2014; Razavi et al., 2012; Sun et al., 2023), and (ii) emulating key dynamic model states or fluxes, then using the resulting emulator outputs (e.g., time series) to cheaply explore parameter-output sensitivities (Bennett et al., 2024; Maxwell et al., 2021). Importantly, this study explicitly focuses on the first strategy, emulation of model performance metrics, which originated primarily within hydrological modeling contexts. This choice greatly reduces the need to run the full hydrological

model iteratively during calibration, substantially lowering  computational expense and enabling scalable optimization for increasingly complex, large-domain hydrology models.*"*

*We also revised this sentence to:* "The large-sample emulator (LSE) approach employs a novel joint training strategy that combines model performance (i.e., response surface) emulation and parameter optimization scheme to estimate parameters jointly across diverse catchments…"*, and there is an explanation of how we trained the emulator: "By training an emulator on a large sample catchment dataset to predict model performance as a function of catchment geo-attributes and parameters…"*

Lines 98-103: But how is the model actually configured? Are these gridded simulations (which seems to be the case based on lines 112-120)?

*Response: As explicitly stated in lines 125-126 (v1); "The SUMMA model configuration adopted a single HRU per GRU, in which the GRU was the entire lumped area of each catchment." we also clarifies lines 112-114 (v1) to remove ambiguity:* "The associated sub-daily forcing, including precipitation, temperature, specific humidity, shortwave and longwave radiation, wind speed, and air pressure, were derived from gridded datasets but spatially aggregated across each basin area, resulting in basin-averaged input time series."

Lines 122-123: "expert judgment and review of model parameterizations (i.e. process algorithms)". This sentence is unclear. What does expert judgement and review entail? In addition, are the model parameters the input parameters to the model or the processes included in the model?  If the latter, maybe it is better to find a different term than "parameters", maybe "configuration"?

*Response: We revised this sentence to explicitly define what "expert judgment" entailed and clarified that the term "parameters" refers explicitly to numerical model inputs rather than processes themselves to clarify explicitly, we have revised lines 122–123 (v1) to:* "expert judgment involving consultation with model developers,  evaluation of previous modeling experiments and sensitivity analyses, and model process algorithms that directly influence runoff generation. These choices include model physics selections, soil and aquifer configuration, spatial and temporal resolution, an *a priori* parameter set and target calibration parameters." *Also, model 'parameters' is a widely used and well understood term in both hydrologic and land modeling, and will be retained.  'Configuration' relates to other modeling choices and is more general.  We retain the terms to abide by convention.*

Section 2.3: This section could benefit from some restructuring (see comments below).

Lines 148-154: Up until this paragraph, the study's subbasin calibration approach (i.e. each subbasin is seen as a single calibration element; not, for example, each grid cell) was unclear to me. This approach could be better introduced in the introduction.

*Response:  We add a clarifying sentence to section 2.1:* "The spatial unit for the calibration experiments is each CAMELS watershed."

Lines 164-169: In addition, this paragraph is better explained in section 2.3.2 (lines 189-199). Perhaps these sections could be restructured as there is a large overlap between the SSE and LSE experiments?

*Response: The introduction contains more general framing material about the study without presenting the details of the method, but now gives a high-level description of the method (eg emulator based, iterative, large-sample) -- see earlier comments -- but a higher level of detail is appropriate in the methods section. We restructure and consolidate by moving figure 2 to section 2.3, with some associated discussion. The main point of this intro part of 2.3 is to give the overview context that there are two main sets of experiments, and give some of the cross-cutting details, e.g. length of run, spinup.*

Lines 164-175: This iterative approach is very similar to traditional calibration approaches except for the speedup offered by the model performance emulator. Moreover, significant numbers of process-based model simulations are still needed, even when considering the generalization opportunities. This trade-off could be better discussed in the discussion.

*Response: This computational trade-off is highlighted in Section 2.3.2 lines 203-207(v1): "The computational demand of the LSE approach was significant; even using an emulator, it still requires conducting a large number of simulations to generate parameter sets based on optimization algorithms, as well as testing them in a computationally expensive LHM. To address this, the number of iterations was minimized while the number of parameter trials per iteration was increased, which we found improved efficiency without sacrificing accuracy."*

*However, we recognize the importance of further discussing this trade-off explicitly in the Discussion section. Therefore, we have added the following paragraph to Section 4 (Discussion and Conclusions):*

"While the LSE strategy still requires a set of process-based model simulations for training, it offers a substantial computational advantage over traditional calibration approaches by drastically reducing the number of required simulations in subsequent iterations. Rather than incurring the cost of repeated full-model evaluations across basins, the emulator enables efficient exploration of the parameter space with far fewer model runs. As described in Section 2.3.2, we further improved efficiency by increasing the number of parameter trials per iteration while reducing the total number of iterations—an approach that maintained accuracy while accelerating convergence. This balance between emulator fidelity and computational cost demonstrates the practicality of the method for large-domain hydrological modeling. Looking ahead, we are optimistic that future enhancements such as adaptive sampling, transfer learning, or cross-domain emulator reuse could further reduce the up-front simulation demand, opening new possibilities for applying this approach to even more complex or higher-resolution modeling systems."

*Also, we disagree with this comment: "This iterative approach is very similar to traditional calibration approaches except for the speedup offered by the model performance emulator." We explain throughout the paper that there are similarities to the emulator-based optimization described in Gong et al (MO-ASMO), and indeed it was a starting point for this work. A major conceptual difference, however, is the large-sample joint training using geo-attributes, a concept which is now common in new ML modeling*

*(as we discuss) but has not been applied to a PB or conceptual model before (in part because it takes a lot of computational effort). The resulting jointly trained emulator offers more than 'speedup' -- it opens the door to potential regionalization, as well as transfer learning from the large sample that we show leads to better performance than can be found in single site training. These are both significant advantages, and very different from the current practice for **PB** models.*

Line 199: This could be a new section, which allows for more detailed description of hyperparameters and cross-validation.

*Response:  This paper has several overarching objectives, the main ones being to present the key findings and outcomes from a new conceptual strategy for calibrating a large-domain complex process model, and to describe the approach & concept. In the two years of development leading to the paper, many sub-focus areas emerged, such as understanding and optimizing the impact of hyper-parameters, which could be the core of entirely different papers. We tested over a dozen different variations, at different stages in the development process. But given that our work is deliverable-directed for US water agencies (i.e., we need to provide a calibrated US wide model by a certain date), we did not have time to do the kind of controlled/extended experiments you would ideally conduct if you were going to present those aspects in a paper.  Also, the results are likely to be highly dependent on the exact application (eg, the model, the size of the catchment collection, the geo-attributes chosen, the computing infrastructure and so on), and along dimensions we had no time to explore. For these reasons, such a broader discussion won't be included here, but reserved for future work should we receive the funding to undertake it. Or it can and probably will be taken up by others who are motivated to do so. We now include the sentence at the end of this paragraph:* "Further discussion of these hyperparameter experiments and workflow development is beyond the scope of this paper, but may be tackled in a subsequent publication after more controlled experimentation."

Lines 420-455: These paragraphs do not discuss the novel aspects and strengths of this study. They do not have to, but they take up a relatively large portion of the discussion.

*Response:  These paragraphs emphasize important context to understand the overall significance of the approach and its findings.  For instance, we describe how the work overturns a commonly held belief about joint calibration for PB models.  We re-emphasize our original motivations arising from the ML community. We summarize/highlight the key conceptual advance and also the potential value. These paragraphs do, in fact, "discuss the novel aspects and strengths of this study" and as such we feel that they are entirely appropriate for inclusion in the paper.  They may help some readers to a more complete understanding.*

Line 423: "differentiablee": differentiable

*Response: corrected.*

Discussion section: Personally, I would love to see more discussion on the trade-offs between different ML/DL based calibration approaches, and the place of this study's calibration approach among them. In addition, I would like to know if and how this study's calibration approach could be used for other (gridded) hydrological models, and to what extent fewer model simulations (e.g. only iteration 0) could be used to generate the same results.

*Response: Unfortunately, such a discussion, unless it is just brief and speculative, is beyond scope. We did compare the new innovation (LSE) to its major logical benchmark, SSE. Such broader papers, involving techniques that would take our group a long time to set up and use in comparative experiments, or alternatively to organize with other method authors, will likely follow as this study's approach is digested by the community. Already, the approach has been recreated by a colleague at another institution, and it is being compared in a new separate paper (under review in WRR) involving some of the authors (but using a conceptual model, HBV) to both pure ML and differentiable learning models run by another group. We feel that the multiple paragraphs included already to orient this work against the concepts arising in ML, or those used traditionally in hydrology, are enough to place it in context.*

*That said, in response to the reviewer's suggestion, we have added a brief paragraph toward the end of the Discussion and Conclusions section that acknowledges the broader momentum in ML/DL-based calibration methods and highlights the adaptability of our framework to other model structures, including gridded implementations:*

"This effort along with Tang et al. (2025) represent initial forays into implementing such a ML-based joint calibration strategy for process models, and each raises as a suite of compelling papers that are beyond the current paper's scope. This paper focuses on introducing, outlining and testing a new large-sample emulator framework, which necessitated substantial dataset, model and workflow development effort, while benchmarking the LSE against a logical baseline, the SSE, and qualitatively comparing it to other related studies using LSTMs, conceptual model and hybrid/differentiable models. We recognize the broader momentum within the ML/DL hydrology community toward methodological intercomparison and refinement, and look forward to undertaking such broader controlled comparisons and studies of methodological choices that were out of scope for this paper. We applied the method to lumped basin-scale PB model configurations for simulating streamflow, but the emulator framework itself is generalizable and could easily be adapted to models with different spatial structures, including gridded domains, levels of complexity, and to multivariate model fluxes and states."

Code and data availability: The code and associated datasets should be made public and cited.

*Response: Our initial draft states that they will be made public, which is our goal, pending acceptance for publication. Accordingly, details of the code and data availability are now being added to the final version.*

**Referee #2 comment for SUMMA HESS preprint**

*We thank Refree #2 for constructive comments. Here we provide our responses to comments from referee #2. The original review comments are in black, and our responses in red, italic font. Blue text was incorporated into the revised paper.*

Farahani et al. evaluate an emulator-based calibration technique, as proposed by Tang et al. (2024. currently as preprint submitted elsewhere) on 600+ CAMELS basins over the US using the SUMMA modelling framework, coupled with the mizuRoute channel routing model. The authors explore the single basin approach (single-site emulator) as a benchmark against the large-sample emulator approach, which integrates static attributes and performance metrics, and provides a basis for large-domain regionalization to unseen basins. The authors have established a comprehensive framework, and their current manuscript is very suitable for publication in HESS after addressing some minor comments listed below. The paper is clearly written and well-referenced.

1. Figures 3,4,5 it's not clear if this is an example of a random gauge/basin or whether it covers the entire dataset shown in Figure 1. In the case of the first one, how are these figures representative? Add this information to the figure captions

*Response: These figures are for all basins, we've revised figures captions. For example revised caption for figure 3 is:* "A scatter density plot of emulator-predicted objective function (OF) values (normalized modified Kling-Gupta Efficiency; NKGE') versus real model OF values for the single-site emulator (SSE) approach across six iterations, aggregated across all basins. Pearson correlation coefficient (R) is shown for each iteration."

*We also revise this sentence in section 3.1 Emulator performance: "Scatter density plots for iterations 1-6, illustrating SSE, LSE calibrations and the LSE_CV experiment,* aggregated across all basins, *are shown in Figs. 3-5, respectively."*

2. Figure 6: can you also show the native KGE values (unscaled into -1 to 1 interval), in the supporting information document? What is the motivation for using 6 iterations?

*Response: To clarify, figure 6 displays KGE' values , ranges from $-\infty$ to 1, not the normalized version (NKGE'). The figure simply shows the range –1 to 1 on the x axis for visualization purposes (and the statistics are still based on all results, not just those in the truncated view). We added a note to figure 6 caption:* "Note that the range of x-axis is set to [–1, 1] for visual clarity; no normalization or scaling has been applied to KGE'."

*In response to your suggestion, we provide the full distribution of KGE' values (including values below –1) across iterations for both SSE and LSE approaches in the reviewer response document (Fig. 1). We chose not to include this figure in the manuscript as we believe it does not add much useful information.*

*The main figures already make it clear that a certain fraction of the scores fall below -1.0, below which point the associated simulations are terrible to the point that their exact score is uninformative.*

[Figure]

***Figure 1.*** *Comparison of calibration performance: CDFs of KGE' for (a) SSE and (b) LSE calibration across all basins over six iterations, with median KGE' values noted in the legend. The blue line represents CDF and median KGE' based on default parameter sets for all 627 basins. Both LSE and SSE approaches start with the same iter-0. This figure displays the full range of KGE' values, including lower-performing outliers, providing context beyond the [–1, 1] x-axis limits used in Fig. 6 of the main text. No normalization or scaling has been applied.*

*As for the choice of 6 iterations: this was based on balancing computational cost with observed improvements in emulator performance and calibration outcomes. In initial testing, we found that performance gains diminished progressively leading up to and beyond iteration 6, with correlation improvements in the emulator plateauing and few additional gains in model skill (as seen in Figures 3–5). Thus, 6 iterations was selected as a stopping point which sufficed to demonstrate the results and discriminate between the SSE and LSE.*

3. It might be useful to plot model CDF performance to some other benchmark performances, which are available from the earlier studies over the used CAMELS basins.

*Response: Most previous CAMELS-based benchmark studies report model performance in terms of NSE, whereas our evaluation uses the modified Kling-Gupta Efficiency (KGE'), which offers a more balanced diagnostic by jointly accounting for bias, variability, and correlation. Perhaps more significantly, other CAMELs studies used different basin selections as well as different time periods for assessment, thus we*

*did not want to suggest that an exact comparison should be made. While these differences makes direct comparison challenging, we have already addressed this point in the Discussion and Conclusions section (Section 4), where we provide more summary-level contextual comparisons to published NSE-based results from studies such as Newman et al. (2017), Feng et al. (2022), and Kratzert et al. (2019). Despite the metric difference, we highlight that our LSE-calibrated SUMMA simulations yield competitive or superior performance in terms of median efficiency scores. We also discuss the limitations and nuances of such cross-study comparisons due to differences in basin selection, validation periods, and calibration strategies. We hope this provides sufficient context for interpreting our results relative to prior work.*

4. To my understanding, yes, SUMMA can simulate river flows. Still, its main strength over simple rainfall-runoff models is that it can also provide more realistic estimates of soil moisture, snow, etc. Could you also check how the model performance after the emulator-based calibration against variables independent from the discharge calibration has changed? Similar way, what Tsai et al, (2021), Nature Communication 12(1):5988 showed either for evapotranspiration of soil moisture?

*Response: We agree that evaluating model performance against independent variables such as evapotranspiration or soil moisture would provide valuable insights. However, such an analysis would require a significant effort to obtain, process, quality control, and integrate additional observational datasets and calibration workflows, which are beyond the scope of this study (i.e., it would be a different study, and definitely a useful one). This direction indeed aligns with a broader set of evaluation goals that we view as more appropriate for a dedicated future investigation. Our current study is scoped around streamflow calibration and evaluation. That said, we fully agree that assessing how emulator-based calibration impacts performance across other water balance components (e.g., ET, soil moisture, snowpack) is a compelling direction for future research, especially to assess the multi-variate realism of calibrated SUMMA simulations. Whether SUMMA can provide 'more realistic' estimates of such variables in the sense of their representation of observations is an open question; certainly the algorithms in SUMMA are more explicit from a process perspective than a simpler rainfall-runoff model, but the actual modeling outcomes can also depend on other factors (e.g., choices related to structure and inputs). A controlled experiment with these considerations in mind would be worth undertaking.*

5. From the figures, I see some evidence that your methodology improved regarding streamflow. Can you discuss how the model run times difference, using your methodology, with the hypothetical example of full calibration of your SUMMA runs (without an emulator)?

*Response: While the exact number of simulations required for a traditional calibration of SUMMA across 627 basins depends on the specific algorithm and stopping criteria, published studies commonly report using between **1,000 to 10,000 parameter trials per basin**. For example, Tolson and Shoemaker (2007)*

*used up to 10,000 DDS iterations per basin; Rakovec et al. (2019) tested DDS with a 1,000-iteration budget and SCE with a 10,000 limit; Cuntz et al. (2015) reported convergence of SCE between 16,000–18,000 iterations; and Spieler et al. (2020) applied a maximum of 10,000 trials. In several cases, all model calibration runs were repeated 20 times with different randomly generated initial parameter sets to obtain robust distributions. These methods, after application, would still require an additional step to create a method (presumably similarity analysis and statistical transfer) if the goal following calibration were regionalization.  Scaling such approaches to a large-domain application like ours would result in **several million SUMMA simulations**, which would be computationally prohibitive. In contrast, our LSE-based approach used **1,000 model runs per basin**, while also enabling **joint calibration and regionalization** across the domain.  A forthcoming paper (in preparation) will demonstrate this 'training basin to full domain' transfer, which is the next stage of this work, while in this one, we sought to examine and quantify its potential through the spatial cross-validation experiment.*

6. Figure 1: polylines of the river should be overlayed over state boundaries. Currently, rivers are ending up in the middle of nowhere, not flowing towards the sea. Why are the colours of lakes different from rivers?

*Response: We appreciate the reviewer's keen observation. In response, we have simplified Figure 1 to better focus on its intended purpose: illustrating the spatial distribution of CAMELS basins and the selected headwater basins. To avoid confusion from misaligned or stylized river polylines—which were based on generalized cartographic features and not hydrologically accurate flowlines—we have removed rivers from the figure entirely. We also adjusted the lake styling and added a legend for clarity. We believe this simplification supports the figure's main purpose without introducing potentially misleading or distracting cartographic elements.*

**Referee #3 comment for SUMMA HESS preprint**

*We thank Refree #3 for constructive comments. Here we provide our responses to comments from referee #3. The original review comments are in black, and our responses in red, italic font. Blue text was incorporated into the revised paper.*

This manuscript, \*\*"Calibrating a large-domain land/hydrology process model in the age of AI: the SUMMA CAMELS experiments"\*\* proposes an emulator-guided calibration framework for the SUMMA + mizuRoute process model across 627 CAMELS basins. A Large-Sample Emulator (LSE) is trained jointly on all basins, contrasted with conventional Single-Site Emulators (SSEs). The LSE improves median KGE′ from 0.30 (default) to 0.76 after six iterations and shows encouraging spatial-transfer skills.

The study is timely and valuable; nonetheless, several issues—some echoed by the other two referees—should be addressed before publication.

**Major Comments**
* * *
1. \*\*Clarify the Emulator Novelty Up Front\*\*

The abstract, title and early sections buried the key innovation—the joint, large-sample performance \*\*emulator\*\* and its iterative coupling with a genetic algorithm.

\* I suggest rephrasing the title to reflect the emulator angle (e.g., "… \*\*emulator-based\*\* calibration of a large-domain PB model").

\* In the abstract and Introduction, define \*\*LSE\*\*, \*\*SSE\*\*, and the six-iteration loop in plain language before diving into the SUMMA/CAMELS background.

\* The abstract statement that the LSE "yields comparable performance to traditional individual basin calibration while enabling regionalisation" is vague; provide the actual median KGE′ values for calibration, temporal validation and spatial CV.

\* Clarify that "regionalisation" refers to parameter transfer, not streamflow prediction alone.

*Response: We agree that the novelty of the joint emulator strategy should be clear upfront. We now add the word 'emulator' into the title, as in '... the SUMMA CAMELS emulator experiments'.*

*We've revised the abstract and introduction to explicitly highlight the novelty: joint training across basins, iterative coupling with genetic algorithms, and the use of geo-attributes for parameter regionalization. We used "parameter regionalization" which refers specifically to parameter transfer, enhancing reader comprehension. Regarding definitions -- we note that an Abstract is necessarily a concise summary of a paper, and cannot invest many words in detailed definitions (all of which are of course contained in the paper). Large-sample and single-site should be relatively intuitive to readers of this type of paper.*

*We choose not to cite any specific KGE values in the abstract, as this makes the #s prone to being taken out of context and lacking meaningful study details -- specific KGE results are best understood when those are also being digested by the reader. It also avoids promoting a reductive hyper-focus (e.g., the current KGE wars) on single #s that would presumably be different if any one of those methodological details in the paper were changed (eg sample sizes).*

2. **Quantification of Uncertainty**

The manuscript frequently acknowledges uncertainty but provides no explicit confidence intervals on the calibrated parameter sets, emulator predictions, or KGE′ skill metrics. Credible intervals from the random-forest emulator or an ensemble of emulator-derived parameter sets (which the authors briefly mention experimenting with) would strengthen conclusions about robustness, particularly in the LSE_CV experiment, where median KGE′ remains modest (0.43).

*Response: It is not a bad suggestion, but out of scope for the paper. We chose not to include such additional CI details for several reasons. First, the paper is already long and dense as it must present a new and complex concept (with the emulator, the optimization, the workflow) and also the details of the model implementation and associated datasets. Adding another set of methods (such as the CI calculation) would make it even longer and difficult to read and track the narrative. Even in its long form (with 16 plots), we had to omit a number of supporting, interesting, side experiments in the interest of achieving a digestible length. Second, this is one of the first papers outlining this new strategy, which we note prompted many interesting questions that remained unanswered (despite over a year of work) about how best to optimize the implementation for performance and transferability; thus we feel it is premature to comprehensively evaluate robustness. At this stage, the paper demonstrates the LSE's clear potential (which alone is a significant scientific contribution), while leaving room for future papers to refine and optimize the LSE strategy and explore tradeoffs in methodological choices and the associated robustness. This is underscored by the middling score of the spatial transfer from the current implementation (KGE 0.43 -- not terrible, but not great), which suggests that refinements are needed before delving too deeply into more elaborate statistical analysis. Third, we do include an indirect indication of robustness in Figure B3, which shows the post-facto maximum of the emulator predicted scores; together with the median score, this conveys the level of variability in the emulator predicted results (and their potential KGE CDFs). More formal analysis is reserved for a future paper in which the LSE is more mature and the paper's objective is to drill down into myriad methodological choices.*

3. **Over-fitting and Generalisation**

The LSE's temporal validation median KGE′ drops from 0.76 to 0.69, while the SSE drops only from 0.69 to 0.65. This suggests that the LSE may over-fit the calibration period. Please provide additional

diagnostics (e.g., split-sample NSE bias, out-of-bag error, emulator feature importance) to demonstrate that the gains are not primarily sequence-specific.

*Response: We know that the gains are not 'primarily' sequence specific because substantial gains are retained in temporal validation. Yet we do in fact expect that the gains in calibration are at least partly sequence specific, together with a number of other potential artifacts of a likely overtraining associated with our first choice of emulator and associated hyperparameters. We would now choose these differently, after more experimentation. We feel that we address this issue adequately and within scope with the following sentence and the end of section 3.3: The LSE-calibrated parameters led to slightly greater loss in skill in validation than did the SSE-calibrated parameters, and this effect was pronounced in the more challenging calibration regions noted above. Although the cause of this effect is unclear, a likely culprit is overtraining -- i.e., that the LSE harnesses more sequence-specific information than the SSE to gain a stronger calibration and validation performance.*

4. **Computational Footprint vs. Practical Benefit**

The LSE still requires ~250,000 initial simulations and another 62,700 per iteration, plus HPC resources. A more precise cost-benefit analysis (CPU hours or carbon cost per 0.01 KGE′ gain) would help readers judge whether the approach is "scalable" for agencies without national supercomputers.

*Response: CPU-hours (and carbon cost) are system dependent and we felt that readers would better estimate their own CPU hour cost by knowing the details of the runs (ie, #, length, iterations) rather than the cost on the Derecho & Casper machines at NCAR. The scalability of the approach arises more in the regionalization context than in the calibration context, and we do not claim that it is an inexpensive approach, with the exception that the use of the trained emulator to search the parameter space clearly provides a huge efficiency. In the calibration context, the improved performance of the LSE may be the more compelling argument for its use. In the regionalization context, the ability to jointly train a single emulator to predict an ensemble of 'pre-trained' parameters for uncalibrated locations, without the training of a subsequent statistical method needed to further transfer parameters. A newly submitted paper from some of the co-authors delves into this scalability aspect further: Global calibration and locally-informed regionalization of hydrological model parameters using AI-based large-sample emulators, by G. Tang, A. Wood, Y. Song, C. Shen, now in review with WRR.*

5. **Benchmarking Against Simpler or Purely ML Models**

The Discussion notes that deep-learning LSTMs achieve higher PUB accuracy, but no direct comparison is presented. If I am not mistaken, Large-sample LSTM benchmarks (e.g., Nearing et al. 2021; Kratzert et al. 2019) routinely reach NSE ≈ 0.8 on CAMELS. Including the CAMELS LSTM benchmark in Figures

6–10 would contextualise how much of the gain derives from better calibration versus model structural strength. For example, they would be far more informative if the LSE/SSE CDFs were plotted alongside:

* the CAMELS LSTM baseline (native KGE or NSE);

* a simple PUB similarity transfer model.

*Response: We noted that we did not standardize our basin collection or validation periods to match those used in the LSTM papers, and it is beyond scope to re-do that aspect and also to set up a simple PUB similarity transfer approach. As noted previously, this paper is already long and adding a new section on a methodology for a similarity transfer method would add complexity and length to the paper at the risk of forcing removal (for length) of the more fundamental and important contributions (i.e., introduction, description and demonstration of the new LSE method). In addition, with the LSTM benchmarks very likely if not clearly exceeding those achieved by the LSE approach with a complex process model, having a more controlled comparison # for the LSTMs would not change this assertion. The paper duly acknowledged in section 4 that the qualitative references "are not controlled comparisons with this one or each other, but nonetheless provide useful context."*

*As with this reviewer's other suggestions, this is a great topic for another paper, and is among those we have even partially outlined ourselves, but did not fit into this paper. Lastly, the paper does benchmark the LSE approach against a logical baseline (the SSE), which is geared toward isolating and characterizing one major and significant advantage of the method.*

This situates the PB + emulator approach within the rapidly evolving "Nearing challenge" landscape without turning the paper into an LSTM comparison. Additionally, the conclusion should temper the statement that complex PB models can now "compete" with simpler models until a direct benchmark is presented.

*Response: Direct benchmarking against LSTM or simpler models involves controlling for significant methodological differences (e.g., NSE vs. KGE, different calibration objectives and periods). We already address these comparisons qualitatively in our Discussion. Note, we do not state (as the reviewer suggests) that 'complex PB models can now "compete" with simpler models', but rather that "**we hope that these findings will update conventional wisdom about the ability of complex PB LHMs to compete with simpler conceptual models in performance, given that our temporal validation across hundreds of basins is on par with that of other published CAMELS-based conceptual modeling studies"**. It is true that our temporal validation results are indeed 'on par' with other published studies, thus this already-tempered assertion is supported by findings of the paper.*

6. **Input-Data and Structural Error Treatment**

Calibration performance is inevitably conflated with forcing and observation errors. The manuscript should discuss how biases in EM-Earth precipitation or ERA5-Land meteorology propagate through the emulator and whether bias-corrected forcings would change conclusions.

*Response: The reviewer may be confusing this modeling work with the different context of land/hydrology modeling of climate change scenarios, or use in prediction applications, in which the forcings are often derived from GCMs/ESMs or NWP climate and atmospheric models, and must be bias-corrected (or downscaled) to reduce input forcing errors. In this case, EM-Earth and ERA5-Land are based on in situ station analyses and thus would actually be the 'ground-truth' dataset used in bias correction. It's true that biases in forcings (and any other input) affect model performance, hence possibly limit both emulator and model accuracy. But this topic is beyond the scope of this study, perhaps one that could be taken up in sensitivity analysis work for a different type of paper.*

7.**Hyper-parameters and Cross-Validation**

Please add a supplementary table listing GA population size, RF tree depth, train/test split, and stopping criteria. These settings are not reported in the Methods/Calibration Framework section, and readers may need the baseline settings to replicate results.

*Response: We have added a supplementary table (Table A4) in the appendix summarizing the key hyperparameters used in our calibration framework. Due to time constraints, only a cursory evaluation of the hyperparameters was performed for this study. We recognize that further optimization of these settings, along with the choice of emulator type, is an important avenue for future refinement of the LSE approach.*

8. **Parameter Identifiability**

The paper optimises 15 parameters (Table 1). Yet, no sensitivity or identifiability analysis is offered. Parameters with negligible influence could be fixed, reducing dimensionality and emulator noise. So, I suggest providing a quick Sobol or Morris screening (even on a subset of basins) to show which parameters matter (or any feature importance methods) and fix the rest to reduce emulator noise. Alternatively, easily add a sentence/paragraph in the Discussion on how the emulator could be used to identify unidentifiable parameters. This would help readers understand the emulator's role in parameter identifiability and dimensionality reduction.

*Response: Feature optimization for the emulator (including parameter and attribute screening and selection) is an area that we noted was beyond the scope of this paper and one that is ripe for exploitation in future refinements of the approach. This paper's objective is presenting an innovative new calibration method, requiring the development of a new code base and associated large-sample datasets for efficiently running SUMMA and mizuRoute, and generation of results for an extensive set of experiments*

*with temporal and spatial validation. For this goal, it was sufficient to use an existing and well-demonstrated set of parameters that are effective in calibrating SUMMA streamflow. As noted in the paper, these were developed by co-author Wood in nearly a decade of prior studies calibrating SUMMA and mizuRoute for streamflow, both locally and in large domains. The paper's results clearly show that the SUMMA parameters used do in fact lead to very competitive calibrations.*

*In the related CTSM-focused paper (Tang et al, 2025), we do use a sensitivity analysis technique (PyViscous) to leverage the results of the LHS-based large sample parameter sets from iter-0 for sensitivity-based selection of parameters by region. In later work with that model where we removed this particular aspect versus using a fixed set of parameters everywhere, results actually improved. We don't doubt that a different set of parameters could lead to better SUMMA model and emulator performance, though perhaps only marginally -- but such a study is well beyond the scope and objective of this particular paper.*

*Nonetheless, we now add the following sentence to the first paragraph of section 2.3.1:* "We note that the emulator can also play a role in identifying and selecting optimal parameters for calibration as described in Tang et al. (2025), which provides details on that usage. "

9.**Beyond Streamflow**

As far as I know, SUMMA's strength is multivariate process realism. While recalibration on ET or soil moisture is out of scope for this paper, you may add clarification in the Discussion acknowledging this limitation and outlining how the LSE could be extended.

*Response: SUMMA is not at all unique in this regard, in that most if not all complex PB hydrology and land models strive to simulate multivariate process realism. It is not a 'limitation' that we focused only on streamflow, as that was the stated/core objective of the work and the paper. Other papers can and will focus on multivariate outcomes, with SUMMA and other models, but that topic is not intrinsic to the paper's objective.*

*In the 2nd to last paragraph, we extend a current sentence to include this sentiment:* "We applied the method to lumped basin-scale PB model configurations for simulating streamflow, but the emulator framework itself is generalizable and could easily be adapted to models with different spatial structures, including gridded domains, levels of complexity, and to multivariate model fluxes and states."

10.**Open Code & Data**

HESS now expects publicly archived code. However, the authors promise to release "upon acceptance/publication"; please provide a permanent repository placeholder and cite it.

*Response: To our knowledge, this expectation does not apply before the paper is accepted, hence our statement of availability. For the reviewer's benefit, the private repository is currently here:*

*https://github.com/NCAR/opt_landhydro*. *Now that the paper is on track for publication, we have now added this and further availability information to the paper.*

**Section-Specific Comments**
* * *
**Abstract**

\* I find that the abstract lacks quantitative results.

*Response: True, this is by design. We do not include quantitative results in the abstract because it is an Abstract, necessarily concise and lacking the contextual details to interpret a quantitative result. Also, we do not wish to promote specific numbers (versus an accurate overview-nature qualitative summary interpretation) to avoid encouraging a narrow focus on highly specific and context-dependent statistics, at the expense of a fuller appreciation of the study that produced them.*

**Introduction**

\* The opening paragraph has no reference to support current calibration challenges.

*Response: We add references to this paragraph:* "Traditional single-site calibration approaches that involve tuning model parameters for individual basins can be time-intensive, spatially non-generalizable and computationally costly, which limits their suitability for large-domain (national, continental, global) applications (Shen et al., 2023; Tsai et al., 2021; Herrerra et al., 2022). Because parameter estimation is vulnerable to sampling and input uncertainty and input errors, such basin-specific methods often lead to spatial inconsistencies in parameter estimates, limiting the model's generalizability across broader regions (Wagener and Wheater, 2006)."

\* The literature review on AI hydrology is comprehensive, but the research hypotheses could be stated explicitly. Two basic examples are "H1: An LSE can achieve ≥x.xx KGE′ improvement over SSE; H2: The LSE_CV will outperform SSE regionalisation." So, you may consider adding a few sentences to clarify the hypotheses and how they relate to the objectives.

*Response: While we did not frame our objectives in the form of formal hypotheses (e.g., H1, H2), the manuscript is structured around clear comparative goals: assessing the performance improvement of the LSE over the SSE approach, and evaluating the potential for parameter regionalization using LSE_CV. These aims are introduced early in the paper and revisited throughout the Results and Discussion sections, where both quantitative and qualitative findings are reported.*

*To further clarify our research intent for readers, we have slightly revised the end of the Introduction to explicitly state the key evaluation goals of the study in a manner similar to hypothesis framing, while preserving the flow and structure of the section:*

*"We compare the LSE results with traditional single-site emulator (SSE) calibration, and comment on avenues for further advances in this direction.* This study evaluates whether the LSE framework can improve

model calibration performance over the SSE, and whether the LSE enables effective regionalization of parameters to unseen basins through spatial cross-validation. The following sections describe and discuss the methods and results of a series of experiments with this approach as applied to a large collection of US watersheds"

\* You may consider merging paragraphs on PUB history and ML advances for conciseness.

*Response: Based on the updated structure of the Introduction, we have relocated the paragraph describing the two main emulator strategies. Rather than positioning it between the discussion of AI calibration history and PUB advances, we now place it after the PUB paragraph. This improves the narrative flow by setting up the rationale for our chosen calibration strategy directly before presenting the study's objectives.*

\* Samaniego and his team presented a framework for "large-sample" calibration of global process models in EGU2025. They developed an algorithm to select representative basins for calibration. According to their work, the selection is a matter of some minutes. Then, the calibration of the selected basins is relatively fast. You may consider adding a sentence or two to clarify how this work relates to the current study and how it differs from the approach taken in this paper.

*Response: It is unclear where this work has been published in the literature, and we do not include it here -- this paper was submitted in December 2024 before the newer presentation at EGU in April 2025. We did not see the presentation at EGU, but it sounds like it is more about similarity-based watershed selection and transfer approach, which we have already mentioned. We do recognize prior Samaniego work on the MPR regionalization approaches, which is germane to the current study.*

**Methods**
**Data sources**

\* Provide a table summarising the 627 basins by climate region, basin area and data periods; this will help interpret Figures 7–12. Alternatively, refer to the CAMELS documentation for a summary of the basins or any paper highlighting this point.

*Response: We add this sentence to the end of section 2.1 Study domain:* "A comprehensive summary of the CAMELS basin characteristics is provided in Addor et al. (2017) and is not reproduced here."
We further note that our CAMELS basin datasets associated with this study will be made available after publication.

\* Since you have employed CAMELS data, which covers mainly small to medium size basins with minimum anthropogenic influence, how do you interpret the results for larger basins? In L83, you mention that the LSE is "Such datasets are well-suited for large-scale modelling"; please clarify what you

mean by "larger domains" and how the results may be interpreted for larger basins. How do you see your approach applied to larger basins?

*Response: We agree that our use of the phrase "large-scale modeling" could be misinterpreted as referring to individual large basins, whereas our intention was to emphasize **large-domain modeling**; i.e., modeling over a wide geographic area that encompasses many individual basins. To clarify, we have revised the relevant sentence in Section 2.1 to:*

*Such datasets are well-suited for large-**domain** modeling due to their rich suite …*

*We've added this paragraph to discussion:*

"While our study focuses on small-to-medium basins in the CAMELS dataset, the LSE approach is being designed for application to large domains (regional to continental to global scale). Applying the emulator-guided calibration strategy to such larger regions may require adjustments to account for greater heterogeneity in factors such as spatial scale, dominant processes and land forms, flow routing complexity, meteorological input patterns, among others."

**Calibration Framework**

* Since we are in 2025, why did you choose 1982 to 1989 as the calibration period? The authors should clarify the rationale for this choice, especially since the CAMELS dataset is available (L153-154).

*Response: We selected the 1982–1989 period for calibration because it offers a consistent and complete period of data coverage across most CAMELS basins, which helps ensure a fair comparison of model performance across the domain. We avoided more recent periods to reduce the influence of land use change, data gaps, or major shifts in climate that could confound the calibration. We've clarified this in Section 2.3: "The calibration period spans six water years, from October 1982 to September 1989, with the first year treated as spin-up and excluded from model evaluation.* This period was selected based on its consistent data availability across basins and its use in previous large-sample studies, allowing for comparability and minimizing confounding effects from land use change or climate trends."

* What if the calibration period is longer or even scattered over the entire period? This part is crucial to understanding the results.

*Response: We don't believe the calibration period being fixed versus scattered has a major effect on the results for this study, especially given our focus on comparing calibration strategies across a large sample. In our previous work with CTSM (Tang et al., 2024), we adjusted calibration periods basin by basin due to data gaps and the need for longer spin-up, since CTSM's deep soil layers can take decades to equilibrate. In contrast, SUMMA requires less spin-up, and the CAMELS data supports a uniform six-year period (1982–1989) with high data completeness across basins. Using the same window for all basins ensures comparability and avoids training the emulator on inconsistent calibration inputs, which*

*could introduce biases. In ongoing current work with the SUMMA-LSE, we are using a collection of over 1300 basins for emulator training and do not require the calibration period to be synchronized.*

\* Since you used 27 geo attributes in LSE training, which differs from SSE, how do you interpret the results fairly? Generally, I want to see a more detailed discussion when comparing LSE and SSE.

*Response: The workflow design reflects the intended purposes of each approach: SSE represents traditional single-basin calibration, while LSE is designed for regionalization and generalization across basins by incorporating 27 static geo-attributes. Our goal was not to hold inputs constant, but to evaluate whether including catchment attributes in a large-sample emulator improves calibration outcomes. This construct has been previously applied in various papers including the LSTM work with entity-aware versus non-entity aware ML models.*

*To ensure a fair comparison, both approaches start from the same initial parameter sets, use the same number of trials, and apply the same objective function and optimization structure, as described in Sections 2.3 and 3.1. In the Discussion section, we also clarify that LSE outperforms SSE not simply due to more input data, but because it leverages cross-basin relationships between attributes and model performance. This joint structure is what enables LSE to generalize, which is especially important in large-domain modeling contexts.*

*The paper already provides an extensive discussion comparing LSE and SSE methods and results while fitting into the length of a journal article, and this aspect (attribute use) is well documented in other literature, such as the Kratzert et al 2024 paper that is cited and discussed.*

\* Your second experiment only covers the period 2003-2009. Why? How does your model work for time prior to calibration? Or recent time? I mean, what if you test the model when we have recorded floods or droughts? This is important to understand the model's performance and results.

*Response: The 2003–2009 period was selected for the second (validation) experiment because it provides good data coverage across most CAMELS basins and is temporally independent from the calibration window (1982–1989). Our goal was to evaluate whether the calibrated parameters generalize well to an unseen period under different hydrologic conditions, not to evaluate the model's ability to simulate specific events like floods or droughts. We agree that testing the model's performance during known extreme events is important, but that type of event-based evaluation falls outside the scope of this study and would require a different experimental setup, leading to a different paper. Our temporal and spatial calibration and validation sampling and simulation approaches are conventional and adequate given the scope and intent of this paper, not needing further discussion.*

\* Equation (2) defining NKGE′ would benefit from a short explanation of why a non-linear rescaling was preferred over simply capping extreme negative KGE′ values.

*Response: We used a non-linear transformation of KGE' to NKGE' to prevent extreme negative values from dominating the learning process or skewing the model evaluation. Instead of hard-capping low values, this transformation smoothly compresses large negative KGE' values while preserving their relative ranking. This approach improves emulator training stability and avoids discontinuities in the objective function that could hinder optimization. To clarify this, we've added these sentences after equation (2): "KGE' ranges from − ∞ to 1, while NKGE' normalizes this range to [−1,1], which is necessary to balance the information weight of each basin during training.* This nonlinear rescaling prevents extremely poor-performing trials from dominating the learning process while preserving their rank order. In contrast to a hard cap on low KGE′ values, this smooth rescaling avoids discontinuities in the objective function, improves emulator training stability, and provides a more interpretable optimization surface."

\* L241: What do you mean by "outliers"? Please clarify. Do you mean basins that have low or negative KGE′ values? Or basins that have a large spread of KGE′ values? Please clarify. If so, why did you not drop them? As you mentioned, the dataset has already been benchmarked, and you know that likely, in some regions, none of the hydrological models will work well.

*Response: As explained in Section 2.4.2, by "outliers" we refer to parameter trials (within basins) that yield extremely low KGE' values, typically due to poor model fit from incompatible parameter combinations. These can occur in any basin, even well-performing ones, particularly in early iterations of the optimization process.*

*We chose not to drop poor trials or bad basins, as doing so would bias the training set and reduce the emulator's ability to learn the full range of parameters–performance behavior. While it's true that some basins are more difficult to model, we intentionally retained them because our goal is to develop a generalizable calibration strategy that can perform across diverse basin types. Rather than excluding low-performing trials or basins, we applied the NKGE' transformation to reduce their influence in a smooth and interpretable way. This ensures more stable training without discarding valuable structural variability in the dataset.*

\* L248: Please clarify "which is necessary to balance the information weight of each basin during training" regarding LSE training.

*Response: . We clarify that "balancing the information weight of each basin" refers to the need to avoid biasing the emulator training toward basins with extremely low (unbounded) KGE' values. Because KGE' spans from –∞ to 1, basins or trials with very poor performance can dominate the objective space and*

*distort the emulator's learning. By rescaling the KGE' values to the bounded NKGE' range [−1, 1], each basin contributes more evenly during emulator training, regardless of its absolute KGE' range.*

**Results**

\* Figure 4: Add an interpretation of the wider scattered density emulator predictions in the lower right corner of the plot.

*Response: The wider scatter, particularly in the lower right quadrant of the early iteration plots (e.g., Iteration 1), reflects cases where the emulator overestimates performance, predicting a high NKGE' for parameter sets that actually perform poorly when evaluated in the full model. This behavior is expected in early iterations when the emulator is still learning the structure of the parameter–performance space with limited training data. As more simulations are added through subsequent iterations, the emulator predictions become more accurate, and the scatter narrows significantly (as seen by Iteration 6). We now add a sentence to clarify this in figure 4 caption:* "The lower-right scatter regions in early iterations reflect emulator overestimation, where predicted performance is high but actual model performance is poor. This misalignment diminishes as the emulator improves over successive iterations."

\* L288-290: Was not clear that in ML the more data you have, the better the emulator is?. Here, you need a clarification. Also, since training and testing are not done on the same set of basins, more details are needed to better understand the results.

*Response: We agree that the ML principle "more diverse data improves generalization" is central here. In our LSE_CV setup, although the emulator is not trained on the test basin itself, it is trained on a large, diverse collection of basins, which allows it to learn generalized relationships between geo-attributes, parameters, and performance. This broader training set provides more structure and constraints than a single-basin SSE calibration, helping guide the optimization even in unseen basins.*

*As we show in Figure 4 and discussed in the text, the emulator improves over iterations as it sees more trials, but its strength comes from learning across basins, not just from the volume of trials but from the diversity of geo-hydrologic conditions. We note that this construct of spatial CV in unseen test cases (out of sample spatially or otherwise) is common in the literature for hydrologic model regionalization and in other fields, and likely to be understood by most readers. It has been explained in several places in the paper, and does not need further explanation without adding redundant text to the paper.*

\* Spatial CV: Only one parameter set per test fold is evaluated, yet the authors note that the best set within the 100 trials often differs. Reporting the inter-quartile range across the top n parameter sets would

illustrate potential performance if multiple sets were propagated. So, if you want to highlight this point, please add a sentence or two to clarify this point.

*Response: As noted in Section 3.4 and shown in Appendix Fig. B3, we acknowledge that the best-performing parameter set (in terms of actual model KGE') is not always the one ranked highest by the emulator. The contrast of the median (Fig 11a) and max (Fig B3) illustrate the range of ensemble parameter set results. We briefly evaluated the performance of small ensembles (e.g., top 5–20 emulator-ranked sets) and found that they tend to yield higher median skill than selecting a single set.*

\* Figure 7: It is almost what I expected to see. Could you please add a panel below to show the result for the whole simulation period? I want to see the validation for a longer period.

*Response: In the interest of keeping the length of the paper reasonable, we choose to retain the current illustration of the calibration period, especially as we show the separate temporal validation period in Figure 10. In addition, it would be unhelpful to include the spinup period in the display. The selected results figures should be sufficient to illustrate the performance of the method and its potential, without turning the paper into more of a technical report. Our selection of results does not greatly differ from what is shown in papers with similar aims.*

\* Please provide a table showing which calibration parameter is the most changed in the LSE as the best emulator. Since you brought Table 1, it would be beneficial to have a comparison table showing the difference between LSE and default.

*Response: Because some parameters have uniform default values across basins and others vary, we summarize both types accordingly in a new table (Table A3 in the Supplement). For each parameter, we report the default value (or default median if spatially variable), the median of the LSE-calibrated values, the percent change from the default, and the min–max range across all basins. This provides a domain-level perspective on which parameters were most affected by the calibration process.*

*We also clarified that* "default values shown in Table 1 are taken from a representative basin (e.g., Basin 05120500) for reference. Some default parameter values (e.g., soil or vegetation-related) vary across basins based on local attributes, and the values in this table may not be globally consistent across the domain."

\* Describe the value of having more iterations after the third one when the model performance is not significantly improved (for example, Figure 9).

*Response: Since the calibration experiments were conducted over six iterations, we evaluated temporal validation across the same six iterations for consistency. Figure 9 reflects this design, allowing us to assess whether performance gains achieved during calibration persist during validation and whether overfitting or convergence behaviors emerge in later stages. While median improvements plateau, continuing through iteration 6 provides additional confidence in the robustness and stability of the*

*calibrated parameters. In some basins, validation scores do improve modestly beyond iteration 3, suggesting that late-stage gains, while small in aggregate, can still be meaningful for specific cases.*

\* It is interesting to see in Figure 10 panels (b) and (d) that SSE shows a lower difference than LSE. Please provide a discussion on this point. Why is that?

*Response: We agree that panels (b) and (d) in Figure 10 show that the SSE exhibits a smaller drop in KGE' between calibration and validation than the LSE. As discussed in the manuscript, we interpret this behavior as a possible sign of mild overtraining in the LSE, which achieves stronger calibration skill but may incorporate more sequence-specific information from the calibration period. In contrast, the SSE, being calibrated only on local basin data, achieves lower peak skill but appears somewhat more stable in time. While this may reflect greater robustness to non-stationarity, the LSE still delivers stronger overall performance and enables efficient large-domain calibration. We observe this slight trade-off in results -- ie it is acknowledged in the Figure 10 discussion. Yet it's also possible that it is not significant, i.e., that with a different training/testing period or different samples of basins, or a different model or different parameters and attributes, or a different form of emulator, or different hyperparameters for the emulator, a different result would be obtained. Thus we do not focus much on this outcome or treat it as a significant finding; rather we note it and offer a thought about the reason. Future papers will delve in greater detail into understanding the exact nature of the temporal and spatial transferability of the LSE-based parameters in an attempt to refine the LSE approach.*

\* Figures B1 and B2: Both samples show overestimation in daily streamflow. However, your monthly streamflow shows a good agreement. Please provide a discussion on this point. What are you looking for in these figures? For hydrological regimes, it is important to see the monthly streamflow. But if you want to work on events, you need to stick to daily (sub-daily) streamflow.

*Response: The time series figures (Appendix Figs. B1 and B2) were included to give a small insight on the type of flow results that are obtained in this experimentation, and to illustrate how calibration can improve both streamflow simulations across both daily and monthly scales for individual basins. With 627 basins worth of results, it is impossible to comprehensively convey to the reader all that we see as researchers in the time and space dimension, but including several timeseries can be instructive and provide an insightful contrast to the multiple summary statistical plots. That said, two simulation timeseries are not enough to make or draw any conclusions from - they are for illustrative purposes which is why they are in the appendix.*

* Please clearly add the temporal resolution of the provided KGE′ in the captions of the results and figures.

*Response: We have clarified in Section 2.4.2 that* "all KGE′ and NKGE′ values reported in the study are computed using daily streamflow simulations." *This ensures consistency across figures and helps readers interpret the results accurately.*

**Discussion**

* The claim that sample size (627) "may be inadequate" for high-quality regionalisation should be justified—e.g., by referencing learning-curve analyses or feature-space coverage metrics.

*Response: We use the words 'may be' rather than a stronger 'is' or 'is likely' because it is an observation based on our experience building the emulator approach, reading papers such as Kratzert et al, 2024 (no single basin), which touched on this question, and watching Google build out a dataset of 30K+ basins for global LSTM training. It is a natural speculation about the possible gains of expanding the basin size (i.e., to represent greater heterogeneity) in our application that we wish to alert readers to. We are unaware of directly relevant learning-curve analyses or feature-space coverage metrics and would lack the scope or available length in this paper to discuss them if they exist -- though that issue of sample size adequacy and potential saturation is one that our team has discussed as a potential follow-on experimental paper. Current work in our group toward expanding CAMELS may offer this opportunity.*

* Elaborate on extending the approach to non-stationary climate scenarios, given that LSE training assumes stationary relationships. Since you mentioned that LSE is "scalable and robust", it is worth reminding readers that its robustness has only been demonstrated under stationary conditions.

*Response: This topic is out of scope, as this paper is not about non-stationary climate scenarios. Another good suggestion for a different paper. We sincerely thank the reviewer for outlining several years of follow-on side studies building on our initial advances in this area and leading to the refinement of the LSE method in its current form (... and we hope that we can find the funding to undertake them all).*

**Minor Comments**
* * *
* You may drop L75-76 after "The following sections ...".

*Response: We appreciate the referee's suggestion but have chosen to retain the sentences "The following sections describe..." because we feel it helps orient and lead readers through the paper. This a stylistic preference and the current authors favor the inclusion of such transition-aiding text elements.*

\* Figure 1 should include a shaded relief of the CONUS to help readers easily see the basins' locations.

*Response: We elect not to add this feature -- our reasoning is that a shaded (presumably grey) relief behind the colored symbols adds visual complexity to the plot that may compete for attention to the colored circle values, while also both diminishing and complicating the contrast with the background -- since in some places the background will be darker or lighter, which makes the colors of the dots appear darker or lighter. The annotation of state boundaries against a simple white background was a choice to make the information of the plot, ie, the color values, stand out more clearly, while giving a geographic reference. We also use a discrete color bar for this reason, versus continuous, to enhance the identifiability of the color-to-value matches. We may try to use the reviewer's suggestion in other contexts, however, such as when in a presentation in which we are presenting one larger plot, versus a multiplot figure.*

\* Vague wording: "That effort (unpublished, led by authors Wood and Mizukami) ... without mizuRoute routing."

*Response: We have revised the paragraph to clarify the nature of the unpublished prior calibration effort led by authors Wood and Mizukami.*

"We note that the 'default' parameter values used here reflected prior study calibration efforts from a site-specific CAMELS-based SUMMA streamflow calibration project conducted by authors Wood and Mizukami (unpublished). The earlier effort used the Dynamically Dimensioned Search (DDS) algorithm (Tolson and Shoemaker, 2007) and calibrated many of the same parameters. However, the work used an earlier version of SUMMA and did not include mizuRoute routing, thus it forms only a baseline reference for our current parameter choices in this study. The prior effort's SUMMA-CAMELS dataset, DDS calibration workflow and parameter selections later contributed to a SUMMA sensitivity study (Van Beusekom et al., 2022) and was published in associated repositories."

*The revised text now more clearly explains that the default parameters were derived from a previous CAMELS-based SUMMA calibration using the DDS algorithm, with an earlier SUMMA version and no mizuRoute routing. We do note that the outcomes of that effort were adopted in another paper.*

\* \*\*Table 1\*\* alignment is off; consider moving lengthy process-importance text into footnotes for readability.

*Response: We corrected the column alignment. To improve clarity, we slightly adjusted the formatting and spacing to ensure the "Process Importance" descriptions are easy to read. We chose to retain the descriptions within the table (rather than move them to footnotes), as they provide immediate context and are helpful for readers scanning parameter definitions.*

\* L159-160: "a large set (400) of parameter combinations" → "a large set of parameter combinations (400)". Similarly, in L170 and L179.

*Response: corrected*

\* L159: You already mentioned "Latin Hypercube Sampling (LHS)" in L130. Similarly, L254 for KGE′.

*Response: corrected*

\* Figure 2: This figure has no high-quality resolution. Please provide a high-quality version.

*Response: such a version will be included in the final publication if accepted.*

\* Figure captions (e.g., Fig. 5) should define acronyms (NKGE′) on first use.

*Response: corrected for all figures.*

\* Line numbers occasionally omit commas in large numbers (e.g., 250800 samples); please format consistently.

*Response: Thanks for pointing this out - we were unaware of this. We carefully reviewed the manuscript and confirmed that numbers like '250,800' are already correctly formatted with commas for readability. On final submission we will proofread for such errors.*

\* Typos: "bains" → "basins" in L318; "have broader spread are have" → "have a broader spread and are" in L339.

*Response: corrected*

\* The Prim sign is different in your caption and the text. Please check the sign of the prim.

*Response: corrected*

\* Drop your figures' main title like "Emulator vs Real model OF ...". It is not needed. For all figures. Please bring everything you need in the caption.

*Response: We agree with the referee's suggestion regarding Figures 3–5 and have removed the unnecessary main titles ("Emulator vs Real model OF ..."). However, we retained subtitles within subplots to clearly indicate iteration numbers, as these subtitles substantially aid figure readability and interpretation. All necessary context and details are now fully incorporated within the figure captions.*

\* Figure 6: Drop all median legends**.** It is not needed. You can add the median in the caption. Also, I think "Best parameter set default" is the wrong term; maybe "default parameter set" is better. Please check. Moreover, to better visualise the legend, you can drop redundant words like "Best parameters".

*Response: We have revised Figure 6, 9 and 11: we removed all median legends and included the median KGE′ value in a (#) at the end of the main legend (e.g., "Iter1 (0.64)"). We have also removed redundant words such as "Best parameters" from the legend.*

\* In some figures like Figure 6, the size of labels is not the same. Please check all figures and ensure the labels' size is the same.

*Response: Corrected. We will carefully review all figures to ensure consistent label font sizes and formatting. A revised version with these corrections will be included in the final publication, if accepted.*

\* Figure 10 has no caption, so it is unclear what it is about. Please add a caption and drop titles and subtitles in the figure.

*Response: Figure 10 indeed had a caption, but due to formatting, it was incorrectly placed at the top. We have corrected this by moving the caption to its conventional position below the figure. We also revised the caption to improve clarity. However, we have retained the subtitles within the figure itself, as we believe these subtitles significantly enhance readability by clearly distinguishing and labeling each panel, aiding interpretation.*

"Figure 10. Temporal validation performance (during the independent 2003–2009 period) shown as spatial distribution across the CONUS. Panels illustrate (a) median $KGE'$ values for SSE calibration, (b) difference between validation and calibration median $KGE'$ values for SSE (c) median $KGE'$ values forLSE calibration, and (d) difference between validation and calibration median $KGE'$ values for LSE."

This study showcases a promising path toward large-sample calibration of process-based hydrology models, but addressing the clarity, benchmarking and uncertainty issues above will significantly strengthen its impact and situate it more convincingly within the broader AI-hydrology dialogue. The authors are encouraged to revise the manuscript accordingly, and I look forward to seeing the next version.

*Response: We appreciate the time and effort that the reviewer put into their review, and their impressive attention to details both large and small, and to broader context and questions that arise in numerous places in the manuscript. It sounds trite but it really does help us to improve the paper. We also regret that a number of the suggestions for added discussion and side analyses are beyond scope -- a paper resulting from responding to all of them would be thrice the length of a normal journal article.*

*This work represents the steady effort of over a year by a small team of people under real-world deadlines that did not allow for tracking down the many questions that arose. We have already excluded of our own numerous side analyses on different aspects to maintain a focus on the core objectives of this initial paper on the LSE-SUMMA implementation, which had a heavy list of key elements to include: introducing and describing a complicated new method involving process model implementation, machine learning emulator implementation, a complex (HPC-based) workflow integrating optimization/modeling/emulator training, large-sample dataset curation for the application, as well as conveying its scientific motivations,*

*significance and real-world relevance. As such the paper is currently longer than we prefer, but it contains sufficient methodological specifics and numerous analyses and results descriptions to be reproducible -- indeed, at least one major external modeling research group already has multiple researchers generating copies of the method, based on the preprint, while we seek full publication and release of the code-base. While we know that there are aspects of the work that can and will be refined (which is normal for a new method), we do not believe that there are any methodological errors in the work or contextual errors in the paper, and we believe that its scientific contribution is notable. We look forward to delving into some of the reviewer's questions that were beyond scope for the current paper, some of which we had previously identified and are just beginning to investigate (provided that we can find funding, as always).*